

# Development of a country-wide seismic site-response zonation map for the Netherlands

Janneke van Ginkel[1,2], Elmer Ruigrok[2,3], Jan Stafleu[4], and Rien Herber[1]

[1]Energy and Sustainability Research Institute Groningen, University of Groningen, Nijenborgh 6, 9747 AG Groningen, the Netherlands.
[2]R&D Seismology and Acoustics, Royal Netherlands Meteorological Institute, Utrechtseweg 297, 3731 GA De Bilt, the Netherlands.
[3]Department of Earth Sciences, Utrecht University, Princetonlaan 8a, 3584 CB Utrecht, the Netherlands
[4]TNO - Geological Survey of the Netherlands, Princetonlaan 6, 3584 CB Utrecht, the Netherlands

**Correspondence:** Janneke van Ginkel (j.a.van.ginkel@rug.nl, ORCID-iD 0000-0001-7601-5119)

**Abstract.**

Earthquake site-response is an essential part of seismic hazard assessment, especially in densely populated areas. The shallow geology of the Netherlands consists of a very heterogeneous soft sediment cover, which has a strong effect on seismic wave propagation and in particular on the amplitude of ground shaking, resulting in significant damage on structures despite the

fact that the events are of small magnitude. Even though it is a low-to-moderate seismicity area, the seismic risk cannot be neglected, in particular, because shallow induced earthquakes occur. The aim of this study is to establish a nationwide site-response zonation by using the lithostratigraphy, earthquake- and ambient vibration recordings.

In the first step, we constrain the parameters (velocity contrast and shear-wave velocity) that are indicative of ground-motion amplification in the Groningen area. For this, we combine ambient vibration and earthquake recordings using resp. the

horizontal-to-vertical spectral ratio method (HVSR), borehole empirical transfer functions (ETFs) and amplification factors (AFs). This enables us to define an empirical relationship between measured earthquake amplification from the ETF and AF, and amplification estimated with the HVSR derived from the ambient seismic field. Therewith, we show that the HVSR can be used as a first proxy for amplification.

Subsequently, HVSR curves throughout the Netherlands are estimated. The resulting peak amplitudes largely coincide with

the in-situ lithostratigraphic sequences and the presence of a strong velocity contrast in the near-surface. Next, sediment profiles representing the Dutch shallow subsurface are categorized into five classes, where each class is representing a level of expected amplification. The mean amplification for each class, and its variability, is quantified using 66 sites with measured earthquake amplification (ETF and AF) and 115 sites with HVSR curves.

The site-response (amplification) zonation map for the Netherlands is designed by transforming published geological 3D

grid cell models into the five classes and an AF is assigned to most of the classes. This presented site-response assessment on a national scale is important for a first identification of regions with increased seismic hazard potential, for example at locations with mining or geothermal energy activities.



# 1 Introduction

Local near-surface lithostratigraphic conditions can strongly influence the level of amplification of seismic ground-motion during an earthquake (e.g. Bard et al. (1988); Bard (1998); Bonnefoy-Claudet et al. (2006b, 2009); Borcherdt (1970); Bradley (2012)). Especially near-surface low-velocity sediments overlying stiffer bedrock modify earthquake ground-motions in terms of amplitudes and frequency content, the so-called seismic site-response.

  Site conditions may be retrieved from available global datasets and the ground-shaking estimation is based on ground-motion

prediction equations (Akkar et al., 2014; Bindi et al., 2014). Site-response estimation require detailed geological and geo-technical information of the subsurface, which can be retrieved from in-situ investigations, however, this is a costly procedure. Because of the time and costs involved, there is a lack of site-response investigations covering large areas, while the availability of detailed and uniform ground-motion amplification maps is fundamental for preliminary estimates of damage on buildings (e.g. Falcone et al. (2021); Gallipoli et al. (2020); Bonnefoy-Claudet et al. (2009); Weatherill et al. (2020)). In the present work,

a procedure is developed to obtain an amplification map for the Netherlands which is both detailed and spatially extensive. Key ingredients are a detailed lithostratigraphic model and a plurality of seismic recordings.

  Overall, the shallow geology of the Netherlands consists of a very heterogeneous soft sediment cover, which has a strong effect on seismic wave propagation and in particular on the amplitude of ground shaking. The Netherlands experiences tecton-ically related seismic activity in the southern part of the country, with magnitudes up to 5.8 measured so far (Camelbeeck and

Van Eck, 1994; Houtgast and Van Balen, 2000; Paulssen et al., 1992). Additionally, gas extraction in the northern part of the Netherlands is regularly causing shallow (3 km), low magnitude ($Mw \leq 3.6$ thus far) induced earthquakes (Dost et al., 2017). Over the last decades, an increasing number of induced seismic events stimulated the research on earthquake site-response in the Netherlands.

  Various studies (van Ginkel et al., 2019; Kruiver et al., 2017a, b; Bommer et al., 2017; Noorlandt et al., 2018) undertaken in

the Groningen area concluded that unconsolidated sediments were responsible for significant amplification of seismic waves over a range of frequencies pertinent to engineering interest. Although the local earthquake magnitudes are relatively small, the damage on the houses can be significant. Hence multiple studies (e.g. Rodriguez-Marek et al. (2017); Bommer et al. (2017); Kruiver et al. (2017a); Noorlandt et al. (2018)) were performed on ground-motion modeling including the site amplification factor for the Groningen region, which forms an excellent study area due to the permanently operating borehole seismic network

(G-network). Here, earthquake recordings and ambient noise measurements, together with detailed subsurface information form an elaborate dataset to study wave propagation in the shallow subsurface (van Ginkel et al., 2019). Local earthquake recordings in boreholes over a range of depth levels show that the largest amplification occurs in the top 50 meters of the sedimentary cover, although the entire sediment layer has a thickness of around 800 m in this region. Furthermore, van Ginkel et al. (2019) showed existence of a the correlation between the spatial distribution of microtremor horizontal-to vertical spectral

ratio (HVSR) peak amplitudes and the measured earthquake amplification. This observation is in accordance with e.g. Perron



et al. (2018) and Pilz et al. (2009) who show a comparison of site-response techniques using earthquake data and ambient seismic noise analysis. In Our study, we first select the Groningen borehole network where a detailed information on subsurface lithology, numerous earthquake ground-motion recordings as well as ambient seismic noise recordings are available. From this we extract empirical relationships between seismic wave amplification and different lithostratigraphic conditions, building upon the proxies defined in van Ginkel et al. (2019).

The microtremor HVSR technique is widely used (Fäh et al., 2001; Lachetl and Bard, 1994; Bonnefoy-Claudet et al., 2006a; Albarello and Lunedei, 2013; Molnar et al., 2018; Lunedei and Malischewsky, 2015) as proxy for site-response and seismic zonation studies and was first proposed by Nogoshi and Igarashi (1970) and widespread by Nakamura (1989, 2019). The HVSR is obtained by taking the ratio between the Fourier amplitude spectra of the horizontal and the vertical components of ambient noise vibrations recorded at a single station. The HVSR of the seismic noise presents peaks which are related to the resonances of shear-waves in the top sediment layer. The HVSR peak amplitude cannot be treated as the actual site amplification factor, but can serve as a qualitative estimate (Field and Jacob, 1995; Lachetl and Bard, 1994; Lermo and Chavez-Garcia, 1993). In this study we focus on the second peak ($\geq 1\,\mathrm{Hz}$) in the HVSR curve which represents the shallow interface of soft sediments on top of more consolidated sediments instead of the resonance of the complete sediment layer as discussed in van Ginkel et al. (2020). This second amplification peak has shown to play a more important role for seismic site-response at frequencies relevant to engineering interest. The HVSR method is applied on the Netherlands seismic network to assess site-response based on ambient vibrations.

The Eurocode 8 seismic design of buildings (CEN et al., 2004) describes the effect of characteristics on soil behaviour during an earthquake and the seismic response of buildings. In order to estimate the risk of enhanced site-response, five soil types are provided based on shear-wave velocities and stratigraphic profiles. Soil-type E in Eurocode 8 is essentially characterised by a sharp contrast of a soft layer overlying a stiffer one. However, in our opinion, this single classification for soft sediments is rather limited, especially concerning the wide variety of lithostratigraphic conditions throughout the Netherlands. Therefore, we present an alternative, or extended, classification for ground characteristics designed to specify the large heterogeneity in site conditions that exists within Eurocode 8 ground-type E.

The aim of this work is to design a site-response zonation map for the Netherlands. Rather than using ground-motion prediction equations with generic site amplification factors conditioned on $Vs_{30}$, a national zonation of amplification factors is developed. To this end, we first select the Groningen region to test empirical relationships between measured earthquake amplification and site-response derived from the HVSR estimations. Next, the ambient vibration measurements of the seismic network across the Netherlands are used, necessary to calibrate the amplification (via HVSR) with the local lithostratigraphic conditions. Combining the detailed 3D geological subrsurface models GeoTOP (Stafleu et al., 2011, 2021) and NL3D (Van der Meulen et al., 2013), with a derived classification scheme, a zonation map for the Netherlands is constructed.

The presented site-response zonation map for the Netherlands is especially designed for seismically quiet regions where tectonic seismicity is absent, but with a potential risk of induced seismicity, for example due to mining or geothermal energy activity (Majer et al., 2007; Mena et al., 2013; Mignan et al., 2015). As a result, this map can be implemented in seismic hazard analysis.



## 2 Geological setting and regional seismicity

The Netherlands is positioned at the southeastern rim of the North Sea sedimentary basin. The sediments at the surface are almost entirely Quaternary with the thickest succession (600 m) occurring in the northwest (Zagwijn, 1989; Rondeel et al., 1996; De Gans, 2007). Neogene and older sediments are only exposed in the farthermost east and south of the country, where the edges of the North Sea Basin were uplifted and eroded. The main tectonic feature of the country is the Roer Valley Graben, bounded by the Peel Boundary Fault in the northeast and the Rijen, Veldhoven and Feldbiss Faults in the south and southwest (Figure 1).

The surface geology is mainly characterized by a Holocene coastal barrier and coastal plain in the west and north, and an interior with Pleistocene deposits cut by a Holocene fluvial system (Rondeel et al., 1996). The coastal barrier consists of sandy beach and dune deposits and is up to 10 km wide. It is intersected in the south by the estuary of the rivers Rhine, Meuse and Scheldt, and in the north by the tidal inlets of the Wadden Sea. The coastal plain mainly consists of marine clay as well as peat. Although much of the peat has disappeared because of mining and drainage, thick sequences of peat ($> 6$ m) still occur. The Holocene fluvial deposits of the rivers Rhine and Meuse are characterized by a complex of sandy channel belt systems embedded in flood basin clays. The fluvial channel belts pass downstream into sandy tidal channel systems in the coastal plain.

The Pleistocene interior of the country mainly consists of glacial, eolian and fluvial deposits. Glacial deposits include coarse-grained meltwater sands and tills. Ice-pushed ridges, with heights up to 100 m, occur in the middle and east of the country. Eolian deposits mainly consist of cover sands, and are locally made up by drift sand and inland dunes. In the south and east of the country, sandy channel and clayey flood basin deposits of small rivers occur.

Neogene and older deposits are only exposed in the eastern- and southernmost areas of the country. In the east, these sediments include unconsolidated Paleogene formations as well as Mesozoic limestones, sandstones an shales. In the south, the older deposits comprise unconsolidated Neogene and Paleogene sands and clays, as well as Cretaceous limestones (chalk), sandstones and shales, and Carboniferous sandstones and shales.

### 2.1 Regional seismicity

The Netherlands experiences two types of seismicity; firstly, earthquakes in the south-east are caused by deep tectonic processes and secondly, induced seismicity at shallow depths triggered by exploitation of gas fields (Figure 2). Tectonic seismicity occurs mainly in the Roer Valley Graben (yellow circles, Figure 2) which is part of a larger basin and range system in Western Europe, the Rhine Graben Rift System. At the beginning of the Quaternary, the rate of subsidence in the Roer Valley Graben has significantly increased (Geluk et al., 1995; Houtgast and Van Balen, 2000) and the rift system still shows active extension (Hinzen et al., 2020). The largest earthquake recorded ($Mw$=5.8) in the Netherlands was in Roermond in 1992, due to extensional activity along the Peel Boundary Fault (Paulssen et al., 1992). Gariel et al. (1995) quantified the near-surface amplification based on spectral ratios of aftershocks from the 1992 earthquake in Roermond. They observed great variety in ground-motion amplitudes over different stations which is very likely a site effect of shallow sedimentary deposits.

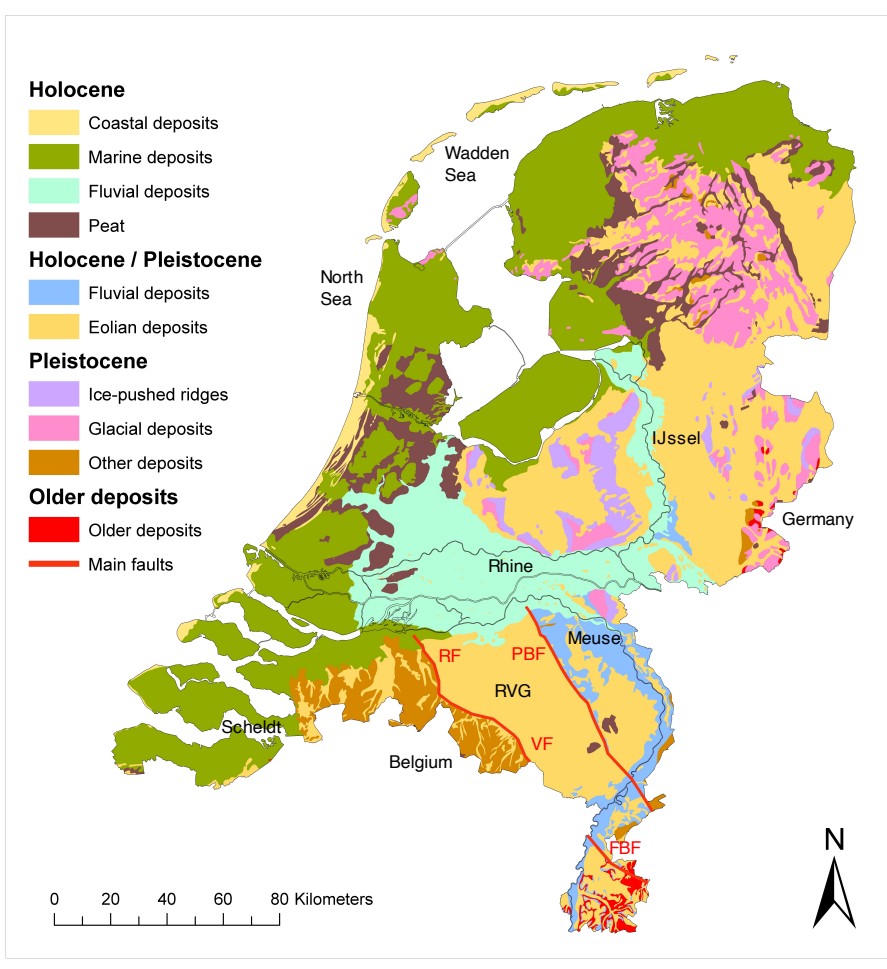

**Figure 1.** Geological map of the Netherlands. Older deposits comprise unconsolidated Neogene and Paleogene deposits as well as Mesozoic and Carboniferous limestones, sandstones and shales. Modified after Schokker (2010). RF = Rijen Fault, VF = Veldhoven Fault, PBF = Peel Boundary Fault, FBF = Feldbiss Fault, RVG = Roer Valley Graben.

Most induced earthquakes in the Netherlands (orange circles, Figure 2) have their epicentre in the Groningen region due to production of the gas field. Here, reservoir compaction due to pressure depletion has reactivated the existing normal fault system that traverses the reservoir layer throughout the whole field. (Buijze et al. (2017); Bourne et al. (2014)). Even though the magnitudes are relatively low (van Thienen-Visser and Breunese, 2015), the damage on buildings in the area is substantial due to shallow hypocenters and amplification on the soft near-surface soils (Bommer et al., 2017; Kruiver et al., 2017a).

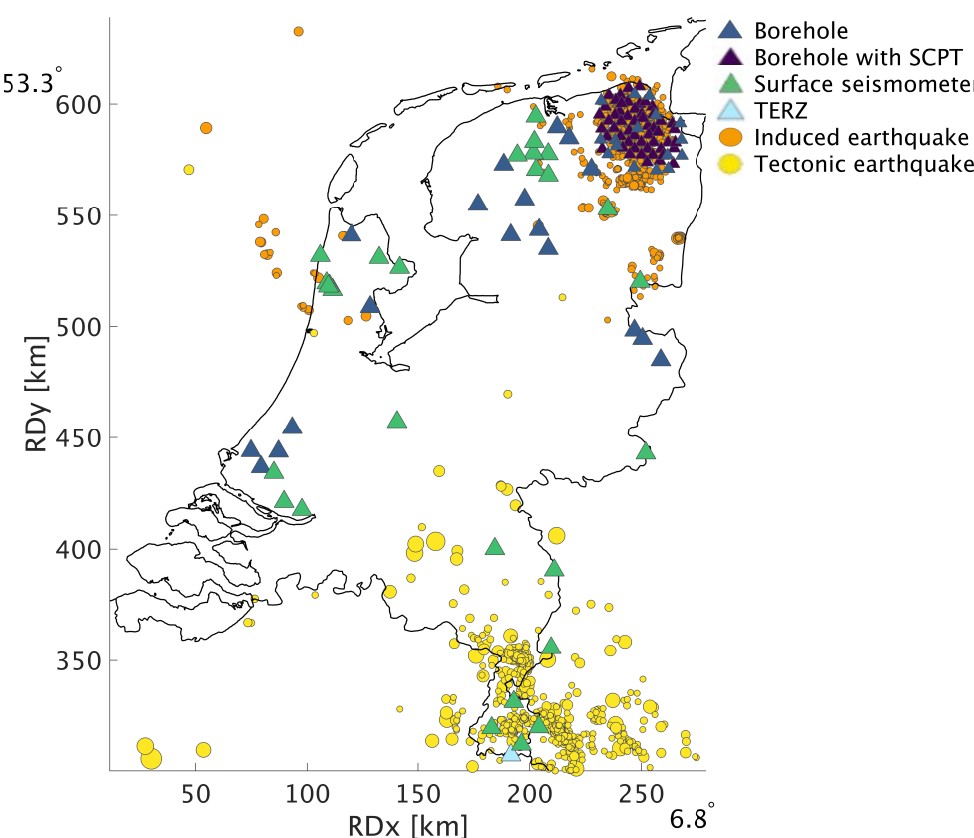

**Figure 2.** Map of the Netherlands depicting all induced ($Mw$ 0.5-3.6, orange) and tectonic ($Mw$ 0.5-5.8, yellow) earthquakes from 1910-2020. The diameter of the circles indicates the earthquake magnitude. The triangles represent the surface location of the borehole stations (blue), borehole stations with SCPT measurement (purple) and single surface seismometers (green). Coordinates are shown within the Dutch National Triangulation Grid (Rijksdriehoekstelsel or RD) and lat/lon coordinates are added in the corners for international referencing.

## 3 Data set


For this study, the seismic network of the Royal Netherlands Meteorological Institute (KNMI, 1993) across the Netherlands is used, consisting of borehole and surface seismometers (Figure 2). The blue and purple triangles represent 88 locations of the borehole network where each station is equipped with three-component, 4.5 Hz seismometers at 50m depth intervals (50, 100, 150, 200 m) and an accelerometer at the surface. The purple triangles indicate boreholes where an S-wave velocity


profile is available from a Seismic Cone Penetration Test (SCPT). The southernmost station is at Terziet (TERZ, light blue triangle in Figure 2). This station consists of a borehole seismometer at 250 m depth and a surface seismometer. The green triangles represent 29 locations of single surface seismometers (accelerometers and broad bands). All seismometers have three





components and are continuously recording the ambient seismic field and, when present, local seismic events. The data is available through the KNMI data portal (http://rdsa.knmi.nl/network/NL/).

In the construction of the site-response map we have made extensive use of the detailed 3D geological subsurface models GeoTOP and NL3D. Both models are developed and maintained by TNO – Geological Survey of the Netherlands (Van der Meulen et al., 2013). GeoTOP schematizes the shallow subsurface of the Netherlands in a regular grid of rectangular blocks (voxels, tiles or 3D grid cells), each measuring 100 by 100 by 0.5 m (x ,y, z) up to a depth of 50 m below ordnance datum (Stafleu et al., 2011, 2021). Each voxel contains multiple properties that describe the geometry of lithostratigraphic units

(formations, members and beds), the spatial variation of lithology and sand grain-size within these units as well as measures of model uncertainty. GeoTOP is publicly available from the TNO's web portal: https://www.dinoloket.nl/en/subsurface-models To date, the GeoTOP model covers about 70% of the country (including inland waters such as the Wadden Sea). For the missing areas we have used the lower-resolution voxel model NL3D, which is available for the entire country (Van der Meulen et al., 2013). NL3D models lithology and sand grain-size classes within the geological units of the layer-based subsurface

model DGM (Gunnink et al., 2013) in voxels measuring 250 by 250 by 1 m (x ,y, z) up to a depth of 50 m below ordnance datum. To determine the depth of bed rock in the shallow subsurface, we consulted the layer-based subsurface models DGM and DGM-deep. These models are also available from the web portal mentioned above. More details on the models GeoTOP and NL3D are given in Appendix B.

## 4    Empirical relationships from the Groningen borehole network

The extensive data set recorded with the Groningen borehole network provides the opportunity to derive empirical relationships between measured amplification in the time and frequency domain, estimated amplification from the ambient noise field and the local lithostratigraphic conditions. Ground-motion amplification is linked to specific subsurface conditions, hence this section elaborates on which of the subsurface parameters mainly influence the level of amplification: the shear-wave velocity and the velocity contrast. First, we define amplification and present results of three different methods to assess it. Next, we compare

the subsurface parameters with the measured amplification to be able to evaluate which parameters are most critical.

### 4.1    Definition of amplification

The majority of site-response studies define the level of soft sediment amplification with respect to the surface seismic response of a nearby outcropping hard rock. Due to the fact that in the Netherlands no representative seismic response on outcropping bedrock is available, we decided to set the reference 'seismological bedrock' at a predefined depth of 200 m. This depth and

corresponding average shear-wave velocity forms the basis from which the site-response and corresponding amplification factors (AFs) are estimated in the next sections. With respect to a reference horizon at depth, waves are also amplified at the Earth's surface due to the free-surface effect. With respect to a hard-rock reference site at the Earth's surface, however, there would be no additional free-surface effect. For this reason, we keep the free-surface effect out of the AF definition.



The Groningen borehole network (G-network) forms a representative resource for the definition of the reference rock param-
eters at 200 m depth. Here, the subsurface is composed of Pleistocene-and Pliocene sediments. At this depth, 95% of the Dutch
subsurface is composed of these sediments at this depth, hence the estimated site-response and corresponding amplification
factors can be applied on a large part of the country. The remaining 5% consist of shallow (<100 m) Triassic, Cretaceous and
locally Carboniferous bedrock, and therefore these locations need to be evaluated separately. Hofman et al. (2017) and (Kruiver
et al., 2017a) determined shear-wave velocities at borehole stations in Groningen. From the velocities found at 200 m depth,
the average is taken, resulting in a reference shear-wave velocity of 500 m/s. At this depth, the density is on average 2.0 kg/m$^3$.

Many studies (e.g., Bommer et al., 2017; Rodriguez-Marek et al., 2017; Borcherdt, 1994) model site-response amplification
factors (AF) for different periods of spectral-accelerations. This study however is empirically driven, taking advantage of the
large amount of high-quality data available. Particle velocity based AFs are derived between the site and the reference horizon
at 200 m, in a frequency range of 1-10 Hz. In the next section we elaborate on the frequency band chosen.

### 4.1.1 Frequency bandwidth

Data processing is applied on a frequency bandpass filter for 1-10 Hz, covering the periods of interest from an earthquake
engineering point of view. Moreover, for these frequencies, the used instrumentation (broadband seismometers, accelerometers
and geophones) have high sensitivity for ground-motion.

Since the majority of amplification is occurring in the top sedimentary layer (van Ginkel et al., 2019), the corresponding
resonance frequencies are covered in the used frequency filter as well. Above 10 Hz the amplitude increases due to soil softening
and resonance is counteracted by an-elastic attenuation and 3D scattering. Furthermore, what exactly happens above 10 Hz is
of little interest since the most energy of the local earthquakes is contained in the frequency band between 1 and 10 Hz. This
is illustrated in Figure 3, which presents the particle-velocity Fourier Amplitude Spectrum of an event recorded on the radial
component in borehole G24.

### 4.2 Amplification Factors

In this study we compute amplification factors (AF) in the time domain from the G-network earthquake recordings. We compute
the AF for each borehole site by taking the ratio of the maximum amplitudes recorded at the surface and the 200 m deep
seismometer. This ratio is taken for both the radial (R) and transverse (T) component and the results are averaged. The amplitude
at the surface was divided by a factor of 2 order to remove the effect of free surface amplification. Next, the AF per borehole is
obtained by repeating the above procedure for all available M>2.0 events and subsequently averaging the values.

We decided to adapt the frequency band and seismometer depth to obtain an AF that is more representative for use on a
national scale than the AF used in the region of Groningen. Hence in this paper, the AF is calculated between the seismometers
at surface and at 200 m depth, for a frequency band of 1-10 Hz. The AF is determined in the time domain and therewith
it provides an average amplification over the applied frequency band. To support the choice for using a single specific AF
frequency band, we calculate the AFs over the Groningen network for several frequency bands (Figure 4). AF values are
highest in the band 1-5 Hz. This is related to the strong resonances in this band. In the 1-10 Hz band the AFs are lower. In the

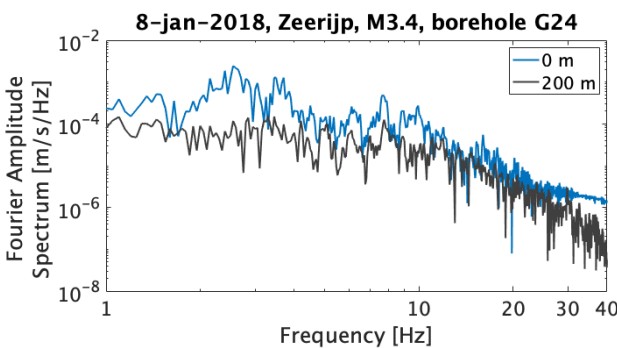

**Figure 3.** Particle velocity Fourier Amplitude Spectrum measured on the radial component for the 200 m (gray) and surface seismometer (blue) for borehole G24 for a time window of 20 s after a local M3.4 earthquake. Borehole G24 has an epicentral distance of 10 km.

5-10 Hz band still considerable earthquake amplitudes are present (Figure 3), whereas less amplification takes place than in the 1-5 Hz band. When the frequency band is extended beyond 10 Hz, the AF-values are not changing much anymore, hence, a representative AF is obtained by limiting the band at 10 Hz.

## 4.3 Empirical Transfer Functions

We compute the empirical transfer functions (ETF) in the frequency domain (in the band 1-10 Hz) from local earthquake motions recorded on the horizontal components of the borehole seismometers to quantify the maximum shear-wave amplification. Here, we apply the same procedure as described in van Ginkel (2021, vertical component, under review) by taking FAS ratios of earthquake records (0-20 s after earthquake-origin-time) at different depth levels in a borehole for local events with M>2. In this study we compute the ETF between the radial component of the seismometers at surface and at 200 m depth ($ETF_{200}$). From the $ETF_{200}$, peak amplitudes are identified which reflect maximum amplification at the corresponding peak frequency. The $ETF_{200}$ derived from the G-network is used for an identification of maximum amplification over the top 200 m that can develop during low magnitude earthquakes. Some example ETF curves are plotted in Figure 5. Additionally, ETF curves for the top 50 m are calculated ($ETF_{50}$). The ETF curves for the different intervals show very similar peak characteristics and peak amplitudes, demonstrating that most amplification develops in the top 50 m of the sediment cover which is supported by the findings of van Ginkel et al. (2019).

## 4.4 H/V spectral ratios from the ambient seismic field

The surface seismometers are continuously recording the ambient seismic field (ASF) and this data is used to estimate horizontal-to-vertical spectral ratios (HVSR). Above 1 Hz the noise field is dominated by anthropogenic sources. Nakamura

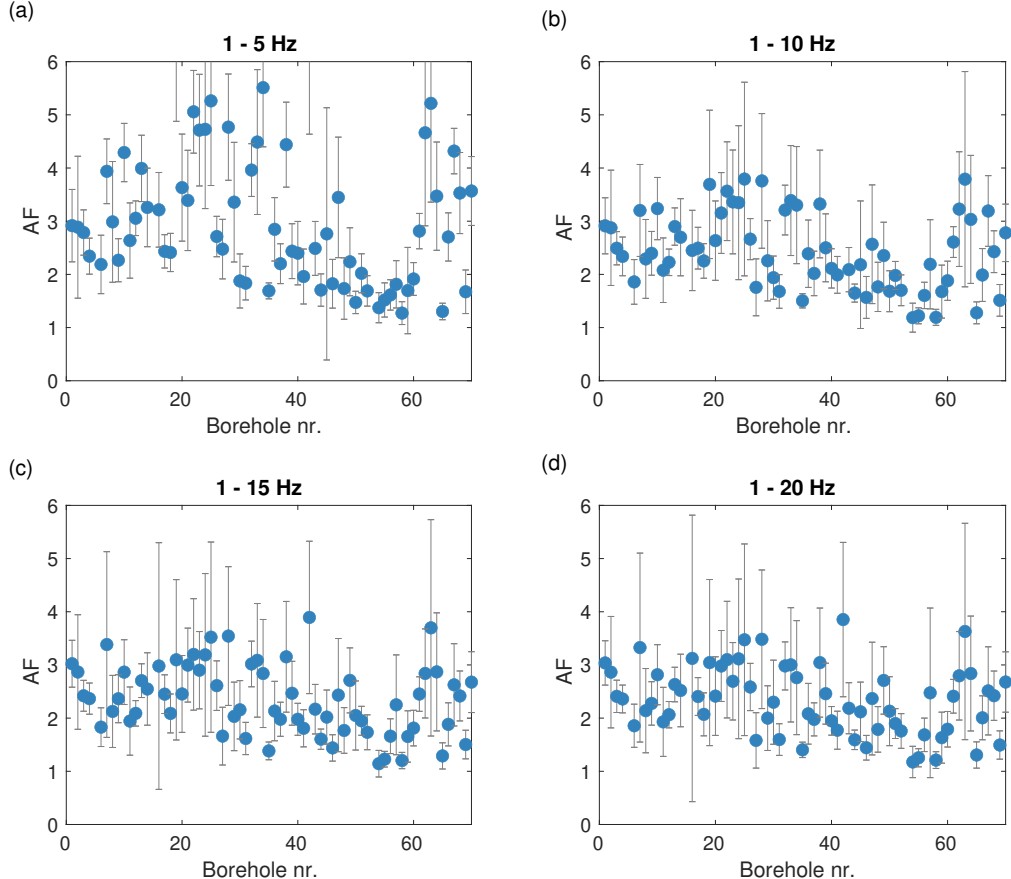

**Figure 4.** Amplification factors (AFs) for a) frequency band 1-5 Hz, b) 1-10 Hz, c) 1-15 Hz and d) 1-20 Hz and corresponding standard deviations (error bars) associated to the averaged AFs over 19 $M > 2.0$ local earthquakes

(1989, 2019) described that the HVSR from ambient noise records is related to the fundamental resonance frequency of the sediment deposits overlying a stiffer bedrock. The details of the method to obtain stable HVSR curves from the ASF in the Groningen borehole network can be found in van Ginkel et al. (2020). In summary, from power spectral densities for each component, the H/V division is performed for each day. Subsequently, a probability density function is computed over one month of H/V ratios and the mean is extracted. This yields a stable HVSR curve that is not much affected by transients like

nearby footsteps or traffic. van Ginkel et al. (2019) presents the details of this methods for frequencies between 1-10 Hz for the Groningen borehole network. Based on the HVSR curve-and peak characteristics, different criteria are defined conformable to the SESAME consensus (Bard, 2002): 1) Clear peaked curves (HVSR amplitude > 4) related to a sharp velocity contrast in the shallow subsurface. 2) HVSR peak amplitude between 2-4, associated to a weak velocity contrast. 3) No distinguishable peak and a flat curve indicate the absence of a velocity contrast in the shallow subsurface. Example HVSRs for these three

criteria are plotted in Figure 5. Its associated peak amplitudes are derived from the mean HVSR curve. The correlation between

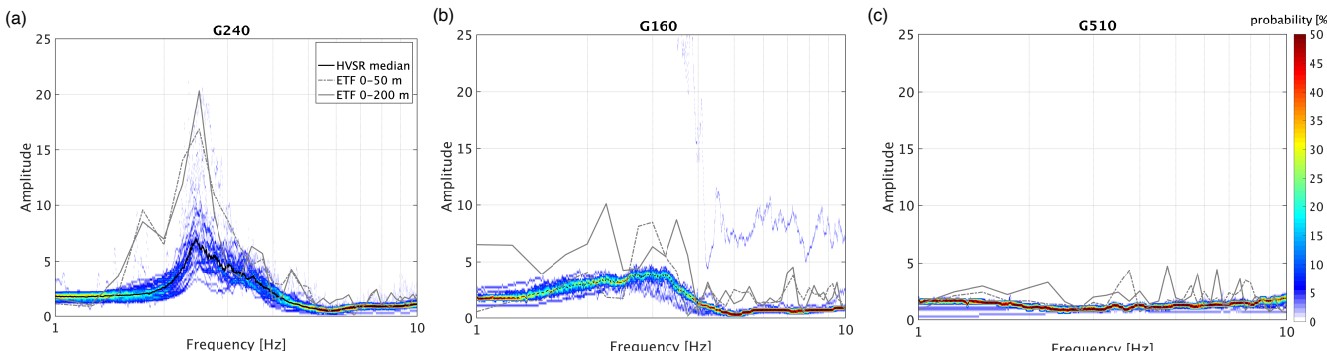

**Figure 5.** Probability density function of one month of daily HVSRs and the mean HVSR (solid black line) and ETF for the borehole seismometer interval of 0-50 m (gray dashed line) and 0-200 m (gray solid line). The selected borehole sites exhibit differences in curve characteristics with a) G24, illustrating clearly peaked curves (>4). b) G16, illustrating medium (2-4) peak amplitudes. c) G51, no pronounced peak.

peaks on the HVSR curves and the presence of a velocity contrast at some depth is stressed in studies from Bonnefoy-Claudet et al. (2008), Konno and Ohmachi (1998) and Lermo and Chavez-Garcia (1993) and this contrast is very likely to amplify the ground-motion.

### 4.5 Amplification parameters

Across the Netherlands, the ASF is continuously recorded on all seismometers, while many locations lack recordings of local earthquakes. Therefore we investigate whether the HVSR can be used as a proxy for amplification as measured by the earthquake-derived ETF and AF. We do this study with the G-network, where all 3 can be measured (HVSR, ETF and AF). Figure 6 displays the correlation between the peak amplitudes of the HVSR and $ETF_{200}$ as well as HVSR and AF. Secondly, we evaluate the subsurface seismic parameters enhancing amplification (Figure 7).

Based on these data points, relationships are defined to be able to estimate amplification factors using HVSR peak amplitudes ($A_0$), amplification factors:

$$AF = 1.49 + 0.87\log(1.12A_0) \tag{1}$$

and maximum amplification as measured by the ETF:

$$ETF = 1.08 + 6.89\log(1.09A_0). \tag{2}$$

Whereas the ETF peak amplitudes represent maximum amplification (at peak frequencies which vary from site to site), the empirical relationship between the HVSR $A_0$ and AF is of most importance for the construction of the site-response map. Therefore, the relationship between the AF and local site conditions is investigated in the following.

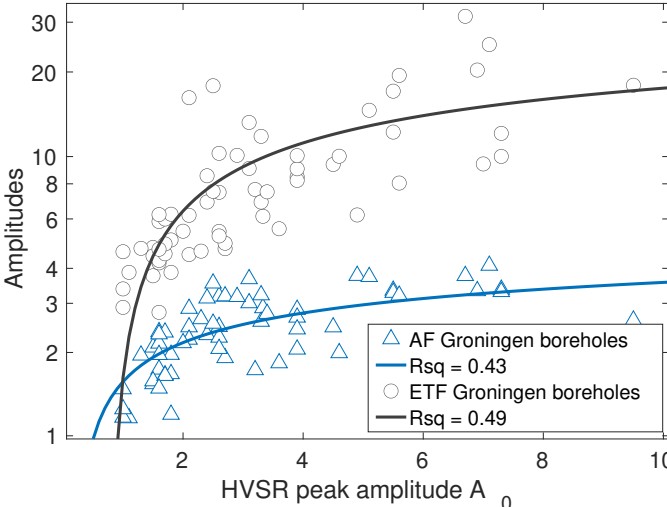

**Figure 6.** Relation between the HVSR peak amplitudes ($A_0$) and the ETF peak amplitudes (gray) and the HVSR peak amplitudes and the AF (blue). The solid line represent the fitting function (Equation 1 and 2 between the HVSR peak amplitudes and the measured amplification from AF and ETF, respectively. $R_{sq}$ (R-squared) represents the coefficient of determination of the fitting. Note the log-scale of the y-axis.

Many ground-motion prediction equations including site-response consider the shear-wave velocity for the top 30 m ($Vs_{30}$) as the main parameter affecting amplification (Akkar et al., 2014; Bindi et al., 2014; Kruiver et al., 2017b; Wills et al., 2000), as well as Eurocode 8 (CEN et al., 2004). However, recent studies (Castellaro et al., 2008; Kokusho and Sato, 2008; Lee and Trifunac, 2010) have drawn attention to the fact that using only $Vs_{30}$ as proxy for site-response is inadequate, because it does not uniquely correlate with amplification, which is defined by several parameters like the depth and degree of the seismic impedance contrast. Hence, the shear-wave velocity ratio between the top and base layer is introduced as a proxy for site amplification by Joyner and Boore (1981) and further explored by Boore (2003). In order to assess the impact of different parameters, first, the AF is fitted, using $A_0 = x_1 + x_2 e^{x_3 Vs}$ as a functional form, with averaged shear-wave velocities over various depth intervals. $Vs_{10}$, $Vs_{20}$ and $Vs_{30}$-values are derived from SCPTs, acquired directly adjacent to 53 borehole sites. Hofman et al. (2017) derived interval velocities determined from the G-network, using seismic interferometry applied to local induced events. The velocities from this reference are used to determine $Vs_{50}$. Secondly, from the SCPT data we derive the depth and size of the velocity contrast (VC) by dividing the shear-wave velocity values for each 1 m interval by the maximum value over the full 30 m is taken as the VC-value. Thereafter the VC-values and their depth are fitted with the AF using $A_0 = x_1 + x_2 log(x_3 VC)$ as a functional form. This procedure is also performed for the relation between the subsurface parameters (Vs, VC) and the HVSR and the results are given in Appendix A.

Figure 7 presents the fit between the AF and the six relevant subsurface parameters. Here, best fit ($R_{sq}$=0.47) is observed between the AF and $Vs_{10}$ and $Vs_{20}$, supporting the findings of Gallipoli and Mucciarelli (2009) by using the $Vs_{10}$ as the main amplification parameter instead of the more common $Vs_{30}$. On the other hand, the correlation between the AF and the VC is less, meaning this parameter is inferior to the AF. Although the fit is relatively poor, a relationship is observable between a





large VC and an elevated AF. By contrast, Figure A1 present a good correlation between VC and HVSR. A large VC-value is leading to resonance in the near-surface, which is expressed in high amplitude peaks of the HVSR. In the next sections, the VC is still considered as an amplification-determining parameter, however, it obtains a smaller weight than the averaged velocity.

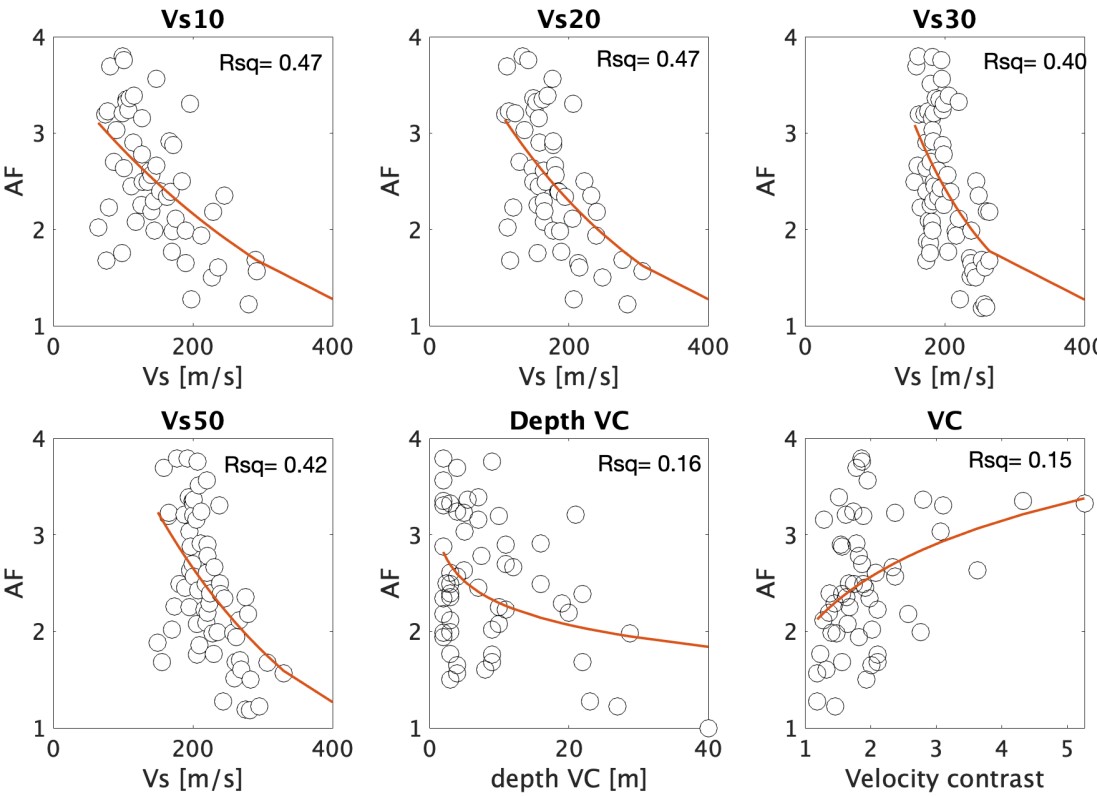

**Figure 7.** Each panel depicts data points in the G-network, the fitted function and corresponding coefficient of determination ($R_{sq}$) between the AF and resp. the $Vs_{10}$, $Vs_{20}$, $Vs_{30}$ and $Vs_{50}$, depth and size of the velocity contrast.

## 5  HVSR estimations throughout the Netherlands

Based on the good relationship between Groningen HVSR peak amplitudes ($A_0$), the ETF and AF (Figure 6), we conclude that the HVSR can be further used as proxy for amplification. Therefore, for all surface seismometers in the Netherlands seismic network, the HVSR curves are estimated following the method described in Section 4.4. Figure 8 displays a selection of representative examples of HVSR curves, plotted as probability density functions and in solid black the mean HVSR. In addition, for boreholes T010, T060 and TERZ, the ETF (red line) is added, calculated from local earthquakes similar to the approach described in Section 4.3. These 16 HVSR curves illustrate the diversity in curve characteristics throughout


the Netherlands. In general, we distinguish the three types of curves as described in Section 4.4. The flat curves with no distinguishable peak (FR040, DRA, T060, T010, WTSB, HRKB, ROLD and BING) are recorded at seismometers on top of outcropping Pleistocene sands (Holocene/Pleistocene eolian and fluvial deposits in Figure 1). Also ALK2 exhibits no peak

amplitude since this seismometer is positioned on dune sands (Holocene coastal deposits in Figure 1), with absence of a strong velocity contrast in the shallow subsurface. Selected examples of HVSR curves exhibiting clear peak amplitudes (NH01,J01, ZH030, FR010, EETW) are located at sites with a distinct velocity contrast between soft Holocene marine sediments overlying Pleistocene sands.

The southernmost part of the Netherlands (Zuid Limburg) has a different lithostratigraphic setting compared to the remainder

of the country. Here, Cretaceous bedrock is either outcropping or situated much less than 100 m deep, resulting in soft rock overlying hard rock. MAME and TERZ are examples of locations with this setting, hence the HVSR curves exhibit a clear peak amplitude. For the TERZ borehole the ETF is displaying similar curve characteristics as the HVSR estimations.

## 6   Zonation map for the Netherlands

In this section, in a few steps, the site-response zonation map for the Netherlands is derived. For this, the country is subdivided

in grid cells. As a result, about 95% of the grid cells is populated with a site-response class with corresponding AF.

### 6.1   Classification scheme

The borehole ETFs confirm that most of the amplification develops in the top 50 m (Figure 5) of the sedimentary cover, which is also discussed in van Ginkel et al. (2019). The top 10 m (Figure 7)is most relevant to explain amplification. According to the lithologic class distribution included in GeoTOP and NL3D, most of the amplification appears in the top of the heterogeneous

and uncompacted sedimentary cover. Beyond 50 m depth, the Quarternary deposits mainly consists of more compacted marine and fluvial sediments. Therefore the sediment classification presented in this section uses the top 50 m with a special focus on the top 10 m. Also the presence of a velocity contrast is used in the classification, as it was shown to have a (albeit weaker) link with amplification (Figures 7 and A1).

In order to account for the influence of local sediment conditions on seismic ground-motion, the European seismic design

of buildings (Eurocode 8; CEN et al., 2004) defines five main soil types, based on lithological description of the sediment column and $Vs_{30}$ (Table 1). Following Convertito et al. (2010) and from the studies by Kruiver et al. (2017a); van Ginkel et al. (2019), the Eurocode 8 classification requires modification caused by the heterogeneous shallow sediment composition and bedrock depth of the Dutch subsurface. Table 1 lists the criteria for the classification division defined for the Netherlands (NL classification). The NL classification is divided into five classes based on the top 200 m lithostratigraphic composition, the

velocity contrast (VC) and average shear-wave velocity for the top 10 m.

For setting up the classification we initially use $A_0$, the peak amplitude from HVSR. The main reason of is that we only have measured AF in Groningen, whereas we have measured $A_0$ for many sites over the Netherlands (all stations depicted on Figure 2). Moreover, we found a clear relationship between $A_0$ and AF (Equation 1).

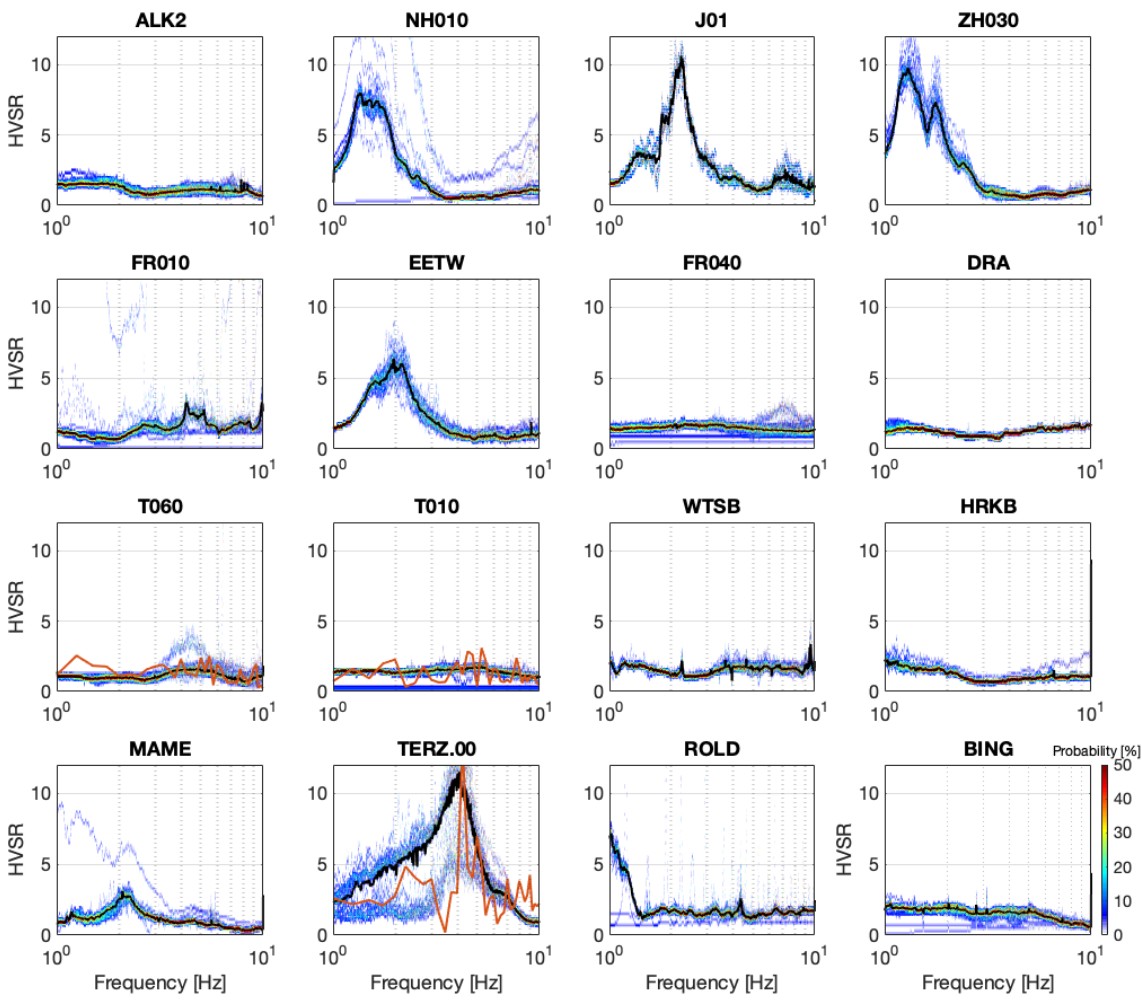

**Figure 8.** Each panel depicts a probability density function from ambient noise HVSR curves for representative stations of the NL-network. The black line represents the mean HVSR. The x-axis is plotted in log-scale. The color bar in the lower right shows the probability range that is valid for all panels. The red line in T06, T010 and TERZ represent the ETF calculated from 10 local earthquakes.

The relationships between $Vs_{10}$, VC and $A_0$ are estimated from lithological conditions as observed in Groningen, where the sites are assigned to Classes II, III and IV. For Classes I and V we have insufficient empirical constraint on $A_0$ and AF. Only sites with bedrock at depths shallower than 100 m fall into Class V. For Class V, the resonance over the complete unconsolidated cover can reach frequencies larger than 1 Hz. Therewith, these sites become distinct from Classes II, III and IV, where such





resonances occur at frequencies <1 Hz. At these smaller frequencies, there is no match with resonance periods of most building types in the Netherlands.

The short lithological description in Table 1 is not sufficient to classify each site. To further aid the classification, representative sediment profiles are obtained (Figure 9) based on the lithologic class sequences of the GeoTOP and NL3D. By grouping the main sediment profiles into the classes, we link the lithostratigraphic conditions to the expected amplification behaviour of the shallow subsurface. The classification is tested and optimized using all the sites with an estimated HVSR curve.

    Next step is to attribute a class to each lithostratigraphy in the GeoTOP and NL3D models.

**Table 1.** Comparison between the Eurocode 8 ground type classification and the sediment classification (NL classification) we present in this paper. The $Vs_{10}$ and velocity contrast (VC) values assigned to each class are based on the amplification relationships presented in section 4 and Appendix A. For class V there is no empirical data available relating $Vs_{10}$ and VC with $A_0$ (HVSR peak amplitude), hence not determined (n.d).

| Eurocode 8 | | | NL classification | | | | |
|---|---|---|---|---|---|---|---|
| Ground type | Description Stratigraphy | Vs30 [m/s] | Sediment class | Description top 200m | $Vs_{10}$ [m/s] | VC | $A_0$ |
| A | Hard rock & rock | >800 | I | Hard rock | >800 | - | - |
| B | Soft rock & very dense soil | 360-800 | II | Stiff sediment | >200 | none or <1.5 | <2 |
| C | Stiff soil | 180-360 | III | Soft sediment on stiff sediment | 100-200 | 1.5-2.0 | 2-4 |
| D | Soft soil | <180 | IV | Very soft sediment on stiff sediment | <100 | >2.0 | >4 |
| E | Special soil | <100 | V | Soft sediment on hard rock (<100 m) | no data | no data | n.d. |

**6.2  Lithology-based classification**

Based on the site-response amplification estimated with the HVSR peak amplitudes at 115 sites, we have categorized each sedimentary profile (Figure 9) into a class. Next step is to substitute GeoTOP and NL3D into these five classes. This geological based method allows the determination of site-response on regional and national scale. Figure 10 gives a general outline of the procedure used to assign the appropriate sediment class to each of the voxel-stacks in GeoTOP and NL3D. A voxel-stack is

the vertical sequence of voxels at a particular (x,y)-location in GeoTOP or NL3D. Details on each of the processing steps are given in Appendix C. Next step is to attribute an amplification factor to each class.

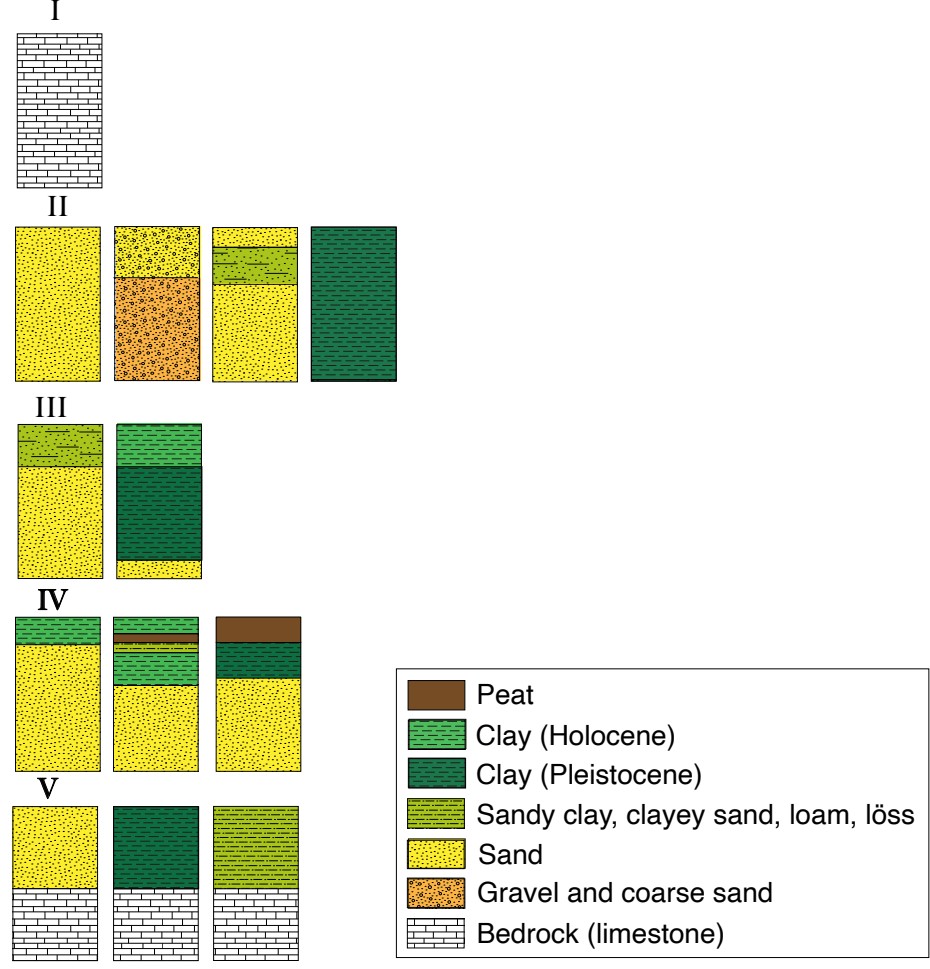

**Figure 9.** Sediment profiles corresponding to the classification presented in Table 1, where the different columns are typical examples of the top 50 m of the Netherlands. The division in classes is based on the shallow subsurface composition related to the expected level of wave amplification during a seismic event.

## 6.3 Amplification factors for the Netherlands

For shake-map implementations or seismic hazard analysis, amplification factors (AF) are usually derived from the $Vs_{30}$ (e.g. Borcherdt (1994)). In this study, we estimate AFs by substituting the HVSR peak amplitudes ($A_0$) for 115 stations throughout the Netherlands into Equation 1. This allows the calculation of nationally applicable AF-values ($AF_{NL}$) assigned to each of the classes presented in Figure 9.

In order to obtain an $AF_{NL}$ for each class, the 115 calculated AFs are plotted against their site sediment class in Figure 11a. For these 115 locations, the sediment classes are manually assigned based on the geological models, SCPT or other geological data available. From the AF distributions, the mean AF-values ($AF_{NL}$) and corresponding standard deviation ($\sigma_{AF}$)



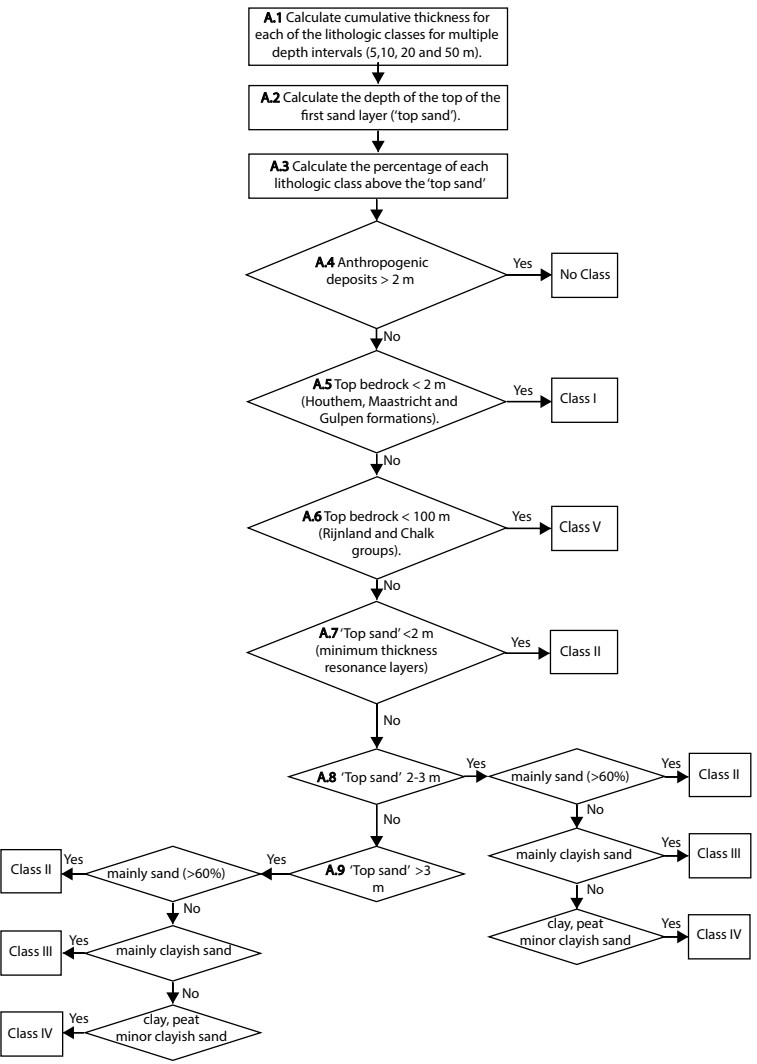

**Figure 10.** General outline of the vertical voxel-stack analysis used to assign the appropriate sediment class into each grid cell of the GeoTOP and NL3D geological models in the construction of the site-response zonation map.

are calculated for each class (Table 2). In Class II there are a number of sites with exactly the same AF of 1.6. These are sites with no distinguishable peak, where $A_0$ is set to 1, which yields, after filling out in Equation 1, AF=1.6.

The $AF_{NL}$-values are valid on a national scale for a frequency range of 1-10 Hz and for reference rock conditions of $Vs$ =500 m/s (Section 4.1). There are no AF values for sites in the farthest south and east of the Netherlands, so these areas fall into Classes I and V. There is too little data to calibrate the corresponding amplification.


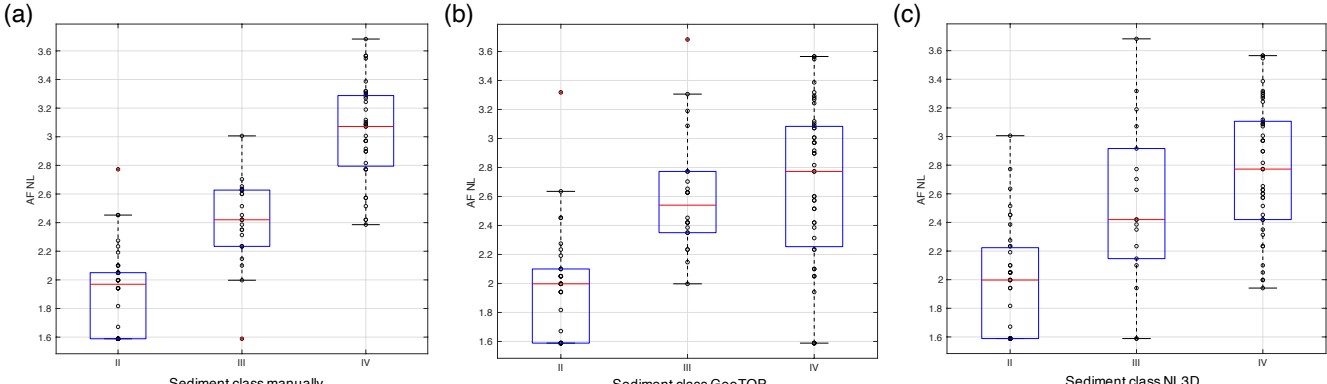

**Figure 11.** Comparison of calculated AF distribution in terms of manual classification (a), automatic classification by GeoTOP (b) and by NL3D (c). The locations where the empirical AF relationship is not valid are eliminated (class I and V). The red central mark indicates the median; the bottom and top edges of the box indicate resp. the 25th and 75th percentiles. The whiskers extend to the most extreme data points and the outliers (1.5x away from the interquartile range) are plotted individually as red circles.

By applying the workflow that we introduced in Section 6.2, automatic classification for the 115 sites is performed based on resp. GeoTOP and NL3D and plotted against the AF (Figure 11b and c). Due to uncertainties in the models (Appendix C), these distributions deviate from the manual classification (Figure 11a). Note that for the manual classification, e.g., SCPT information could be used at 53 sites, which local information is not included in GeoTOP and NL3D. We therefore distinguish two types of uncertainty:

1. $\sigma_{AF}$: this is the variability that comes from the classification. Within the classification, a number of different sites is binned into the same class (Figure 9) although in reality there is still a range of amplification behavior. This variability is approximated with the outcome of the manual classification (Figure 11a), which could be done in great detail.

    2. $\sigma_{mod}$: the geological models are geostatistical models where not all grid cells contain individual lithological data. Hence, there is an uncertainty of the actual lithological succession at each grid cell. The total uncertainty $\sigma_{tot}$ (derived from
Figure 11b,c) can be written as $\sqrt{\sigma_{AF}^2 + \sigma_{mod}^2}$. By additionally averaging over the classes (labeled with subscript $_i$) we find the model uncertainty $\sigma_{mod}$:

$$\sigma_{mod} = \frac{1}{n} \sum_{i=1}^{n} \sqrt{\sigma_{tot,i}^2 - \sigma_{AF,i}^2}. \tag{3}$$

Table 2 lists the mean AF values, the uncertainty in AF ($\sigma_{AF}$) and the uncertainty ($\sigma$) for the GeoTOP and NL3D models.

## 6.4    Site-response zonation map

The workflow presented in Figure 10 results in a class category assigned to each grid cell of the GeoTOP and NL3D models. As a result, we present the national site-response zonation map (Figure 12), were each class characterises a certain level of



**Table 2.** Amplification factors and standard deviations ($\sigma$) for the NL classification. $\sigma_{AF}$ is the uncertainty when a local (HVSR) recording is available. $\sigma$ GeoTOP and $\sigma$ NL3D represents the additional uncertainty associated with the GeoTOP and NL3D models.

| Class | $AF_{NL}$ | $\sigma_{AF}$ | $\sigma$ GeoTOP | $\sigma$ NL3D |
|---|---|---|---|---|
| II | 1.94 | 0.30 | - | - |
| III | 2.4 | 0.28 | 0.32 | 0.34 |
| IV | 3.03 | 0.34 | - | - |

expected site-response amplification. Each class has an $AF_{NL}$ assigned (Table 1). Figure 13 presents four zoom-in panels of the map, each depicting a region of particular interest.

Some areas show a large scatter in classes, which is derived from a large heterogeneity in the near surface as represented in the lithostratigraphic models. Typically, at these places there is large model uncertainty. For example in north-east Noord-Holland (Figure 13a). Here, the Holocene lithological successions are very heteogeneous in terms of clay, peat and clayish sand. This region also exhibits discrepancies between the model's lithological successions and HVSR curve characteristics, for instance with seismometers J01 (Figure 8) and J02. The geological model at these locations presents large portions of clayish sand, resulting in class category III, while the HVSR curves exhibit distinctive, high amplitude peaks, demonstrating local conditions related to class IV.

For larger sedimentary bodies, like the dune area, there is less model uncertainty. Dune sand is identified as class II, and here, the HVSR of the seismometers (e.g. ALK2, Figure 8) deficit any peak due to the absence of an velocity contrast in the near-surface.

Figure 13b covers the "Randstad" region, most densely urbanize part of the Netherlands, where the class is mainly determined as IV. Figure 13c shows the southeastern part. Most of the northern part of this region is Class II due Pleistocene sands reaching the surface. Most of the southern part of this region falls into Class V since the bedrock occurs at depth less than 100 m. A few places with bed rock outcrops fall into Class I.

Since Groningen has been studied in much detail, we also present the site-response zonation for this region (Figure 13d).

## 7 Discussion

The seismic site-response zonation map presented in Figure 12 distinguishes five classes, each of which defining the potential of occurrence of the related site-response. Here, the lithological conditions are collated into zonations (classes) using the classification as shown in Figure 9. In the development of the lithostratigraphically based classification, we used i) HVSR peak amplitudes, ii) the presence of a velocity contrast at depth, iii) shear wave velocities. Amplification factors are assigned to each class. In the following paragraphs we discuss the validity and uncertainties of the classification, the AF distributions, as well as the usage and limitations of the presented map.

Since the ambient noise sources in the frequency band of interest (1-1 Hz) partly have an anthropogenic origin, one should be careful about contamination by local strong noise because it may seriously affect the amplitude of the HVSR as shown in
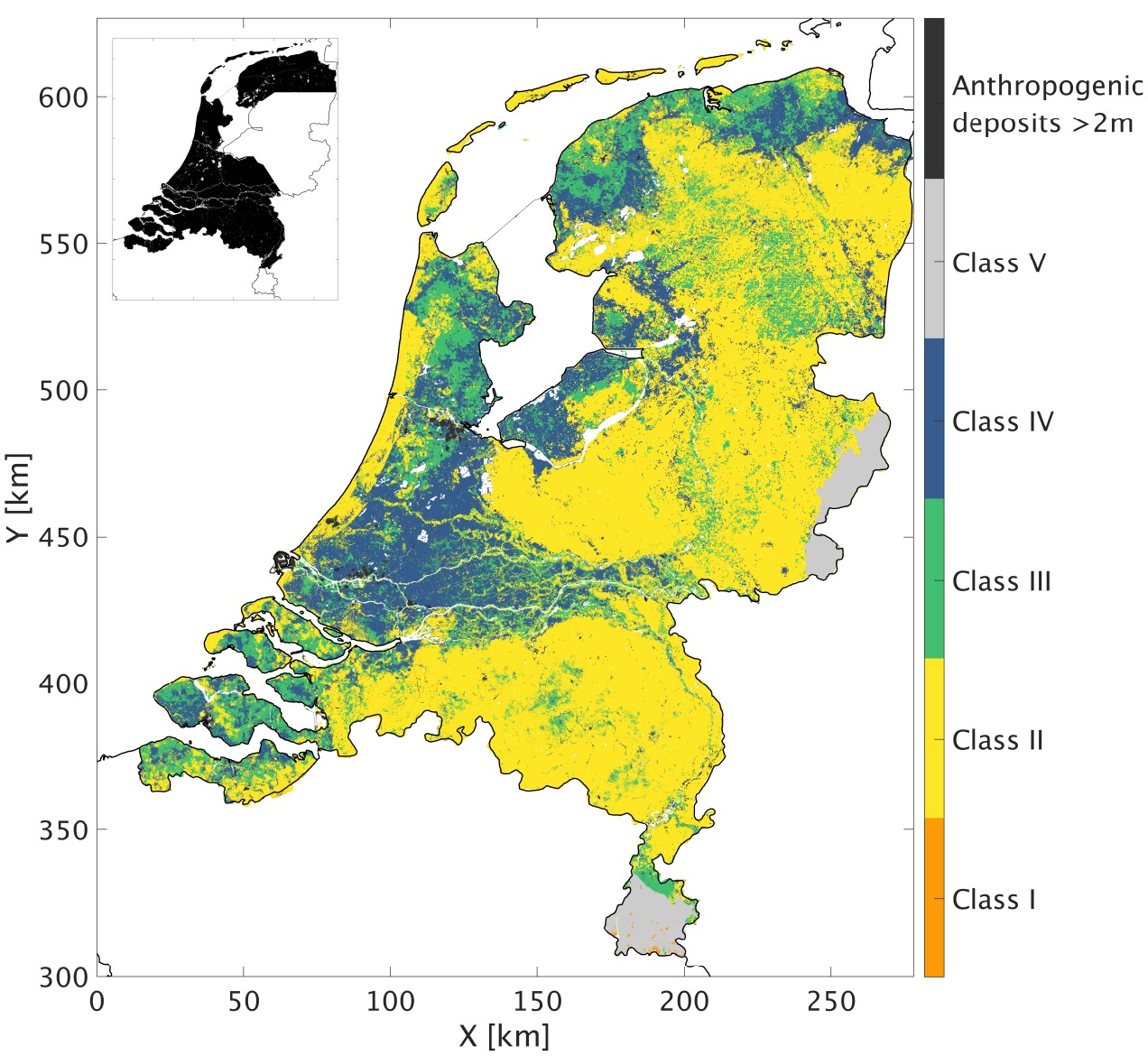

**Figure 12.** Seismic site-response zonation map for the Netherlands designed for low-magnitude induced earthquakes. The GeoTOP model coverage is highlighted in black in the small inset. For the remaining part of the Netherlands, the NL3D model is used as foundation for the classification. The white spots are water bodies. The amplification factors and related uncertainties are presented in Table 2.



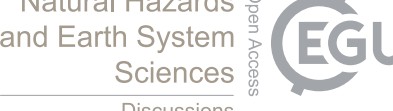
**Figure 13.** Panels highlighting different regions in the site-response zonation map, including the seismometer locations. a) Noord Holland: with a heterogeneous pattern between class III and IV, b) The densely urbanized area of Zuid Holland, the red line indicates the S-N cross-section through the GeoTOP voxel model (Figure C1). c) Limburg: which is in the north a quite homogeneous zone of class II, while the south is dominating by class I and V due to shallow and outcropping bedrock. d) North-east Groningen is added as comparison to other studies performed in that region. No seismometer locations are plotted here because of the high density covering the map.





Guillier et al. (2007); Molnar et al. (2018). We resolved this problem by using large portions (30 days) of noise data to create stable HVSR curves (van Ginkel et al., 2020). It is important to mention the qualitative character of the microtremor HVSR peak

amplitudes which in itself do not directly relate to the amplification of a signal at the surface during an earthquake. However, the microtremor HVSR curve characteristics show major similarities with the measured amplification from earthquake ETFs (Figure 5), but not in terms of absolute values. Therefore an additional fitting-relationship (Equation 1) has been defined, suitable to use the microtremor HVSR peak amplitudes as proxy for amplification. HVSR measurements have proven to be very informative for site-response estimation and remain a valuable input for seismic site-response zonation (Molnar et al.,

2018; Bonnefoy-Claudet et al., 2009).

Considering the difficulty in observing sufficient numbers of earthquake ground-motions in areas that are not seismically active, or where no large seismic networks are available, we resorted to deriving and calibrating a lithology-based classification scheme. We took advantage of the detailed models of Cenozoic lithostratigraphy which are available in the Netherlands. As a consequence, the site-response map (Figure 12) exhibits an which is rather similar to the geological map (Figure 1). We showed

that the use of these models yields additional uncertainty in the determination of the AF (Table 2). This uncertainty of the actual lithostratigraphic profile at a site can be circumvented by a local recording. This may be an HVSR to obtain more certainty on the site effect (Table 1), a cone-penetration test (CPT) to obtain constraints on the lithology, or, better still, a seismic cone penetration test (SCPT) to get a local S-wave velocity profile.

Rodriguez-Marek et al. (2017) defined a site-response model including magnitude and distance dependent linear amplifi-

cation factors ($AF_{Gr}$) for several period intervals for the Groningen region as input for ground-motion prediction equations by Bommer et al. (2017). This site-response model starts from a reference horizon at the interface between the unconsolidated sediments and the stiffer Chalk formation below at around 800-1000 m depth. However, this contrast is both variable in depth and value throughout the Netherlands and therefore not easily applicable as a reference horizon for the purpose of our study. Rodriguez-Marek et al. (2017) presented model-based AFs ($AF_{Gr}$) for several periods in the range of 0.01-1.0 s.

The class-dependent $AF_{NL}$ presented in this paper is defined against a reference rock with a velocity at 500 m/s (which in Groningen is situated at 200 m depth). Therefore the $AF_{Gr}$ cannot be directly be quantitatively correlated to the $AF_{NL}$; this requires a correction which includes the transmission coefficient calculated at the base of the North Sea Group and a damping model. By ignoring the absolute values and comparing both $AFs$ qualitatively, the overall spatial distribution of $AF_{NL}$ in the Groningen region (Figure 13d, in a frequency band 1-10 Hz) corresponds best with $AF_{Gr}$ at a spectral period of 0.01 s (Figure

10; Rodriguez-Marek et al., 2017). This is in line with or findings that AFs do not change much anymore when frequencies above 10 Hz are included (Figure 4).

## 7.1 Usage of the site-response zonation map

The map presented in Figure 12 enables a prediction of site-response after a local earthquake as recommended in the following. It is very important to note that lithological information from geological voxel models is based on spatial interpolation and

aimed at interpretations on regional scale. As a consequence, the presented site-response zonation map is also designed for regional interpretation, but not on individual grid cell scale. Furthermore, at locations with large subsurface heterogeneity, the


interpretation should be handled with care. Additional local investigations like SCPT measurements should be performed at sites of interest in order to assess the site-response in detail.

For the map presented, the uncertainties to keep in mind are: first, the AF distribution along the classes (Figure 11a), and secondly the uncertainty of the geological model used ($\sigma$ GeoTOP and $\sigma$ NL3D, Table 2). The $AF_{NL}$ is designed to be added to an input seismic signal with reference seismic bedrock conditions with a shear wave velocity of 500 m/s. This $AF_{NL}$ is class-dependent and covering only frequencies of 1-10 Hz. Furthermore, the $AF_{NL}$ including the $\sigma_{AF}$ does not reflect the maximum amplification that might occur within a smaller frequency band.

The frequency content of large tectonic-related earthquakes differs from induced tremors. The national AF is based on low-magnitude induced earthquakes and incorporates a frequency range of 1-10 Hz. In case of a strong tectonic earthquake, frequencies below 1 Hz start to play a role and resonances with deeper velocity contrasts (>100 m) which are not reflected in the current $AF_{NL}$ might become important. Also, for very strong ground-motion, which would occur in the epicentral area of large-magnitude tectonic events, non-linearity and distance dependence could become important (Bazzurro* and Cornell, 2004; Kwok et al., 2008). Both effects have not been included in the derivation of the $AF_{NL}$. Moreover, in the country's southern regions, a topographic effect may influence the site-response. It is important to mention that for now these areas are aggregated in Class V and require additional detailed site investigations for site-response assessment.

## 8   Conclusions

In this paper we presented a workflow to create a nationwide site-response zonation, using lithological sequences as proxy for seismic site-response. To that end we first analysed the observed earthquake and ambient seismic field recorded at 69 stations of the Groningen borehole network in order to obtain empirical relationships for amplification. Based on the shallow subsurface resonance frequencies and earthquake amplitude spectra, the earthquake and ambient noise frequency band-pass filtering was applied in the range 1-10 Hz. Derived from the Groningen empirical relationships, we showed that the horizontal-to-vertical spectral ratio (HVSR) approach provides a simple means of determining the amplification potential for most subsurface conditions in the Netherlands. In a second stage, we determined the HVSR curves for additional 46 surface seismometers throughout the Netherlands and calculated the subsequent peak amplitudes. These peak amplitude distributions were related to specific lithological profiles and amplification factors. With the accrued knowledge of amplification potential of different lithological sequences, a classification scheme was designed. This turned out to be a useful tool for translation of the grid cells of the geological models into five classes, and therewith establishing a national site-response zonation map. Most classes have an $AF_{NL}$ assigned, which values can be added to input seismic responses adhering the reference seismic bedrock conditions.

Class I are sites with a hard rock setting. These sites can only be found in the very south and east of the Netherlands. An amplification factor (AF) of 1, meaning no amplification, is assigned to these locations. Class II is associated to sites with stiff sands or Pleistocene clays without strong impedance contrasts in the near surface. One may expect only small amplification at these sites. Class III are sites with relatively soft sediments (clays, sandy clays, löss) overlying stiffer sands, resulting in impedance contrasts in the near surface. Class IV is related mostly to very soft and unconsolidated Holocene clay and peat





successions overlying stiffer sands, forming a strong impedance contrasts. At these sites, the largest amplification occurs. Class V are sites at which the bedrock occurs shallower than 100 m, which is not very common in the Netherlands. For these sites there was insufficient data to assign an amplification factor.

Some limitations exist in this study. The method and map proposed is not applicable to regions with strongly deviating lithological sequences, or for earthquakes with very strong low-frequency (f<1 Hz) shaking.

Finally, it is worth noting that the proposed map could be improved by i) adding new site geotechnical data like SCPTs, ii) including updates and extensions of GeoTOP, iii) including amplification factors derived from new KNMI stations and iv) adding new records of earthquake motions to constrain amplification factors for class V.

## Appendix A:  HVSR amplification parameters

In this appendix, HVSR peak amplitudes ($A_0$) are fitted with the six parameters that influence ground-motion site-response

(Figure A1). Best fit ($R_{sq}$=0.39) is observed between $A_0$ and $Vs_{10}$. Hence, the $Vs_{10}$ is used for further correlation purposes instead of the more common $Vs_{30}$, supporting the findings of Gallipoli and Mucciarelli (2009) by using the $Vs_{10}$ as main amplification parameter. The depth of the first strong velocity contrast (VC, which is defined within the top 50 m) has a poor relation with $A_0$ (Figure A1e). The size of the velocity contrast, however, does have a strong relation with $A_0$ (Figure A1f).

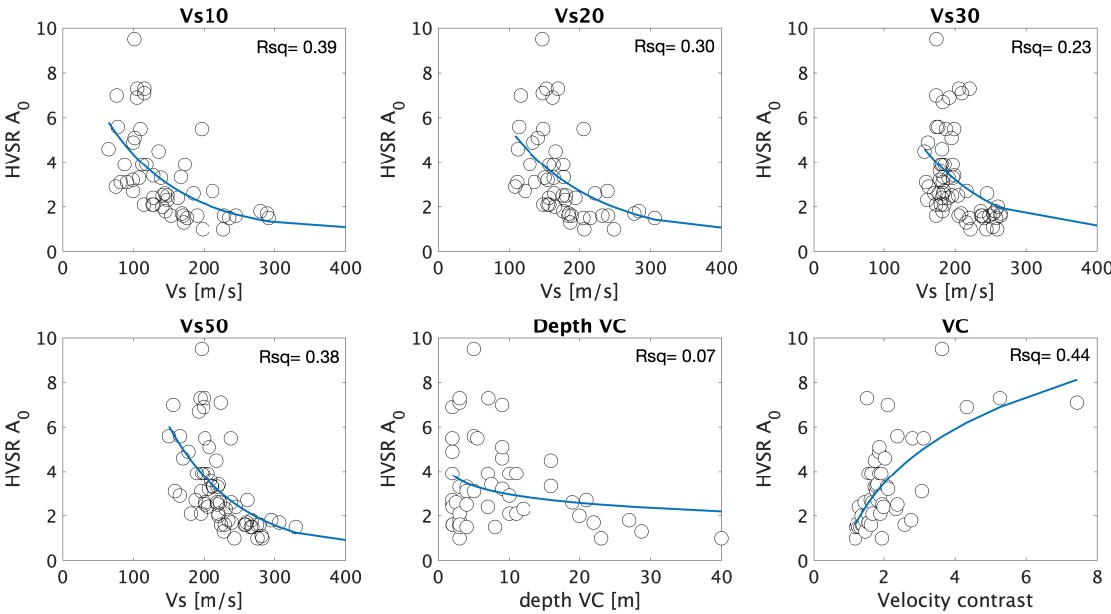

**Figure A1.** Each panel depicts data points of the G-network, fitted function and corresponding coefficient of determination ($R_{sq}$) between the HVSR peak amplitude and resp. the $Vs_{10}$, $Vs_{20}$, $Vs_{30}$ and $Vs_{50}$, depth and size of the velocity contrast (VC).




Compared to individual 1D correlations, a 2D correlation (Figure A2) using both the VC and the $Vs_{10}$ results in an improved

correlation ($R_{sq}$=0.53) and allows to define an empirical relationship for HVSR peak amplitudes ($A_0$) based on these two

parameters:

$$A_0 = -1.29\log(0.01Vs_{10}) + 0.99VC + 1.94 \tag{A1}$$

Furthermore, this equation supports the hypothesis of Joyner and Boore (1981); Boore (2003) that $A_0$ is depending on also

the VC. The motivation for equation A1 is to achieve an amplification equation based on subsurface parameters only. Using

equation A1 an estimate is obtained of $A_0$. Subsequently, Equation 1 can be used to obtain an estimate of the amplification

factor.

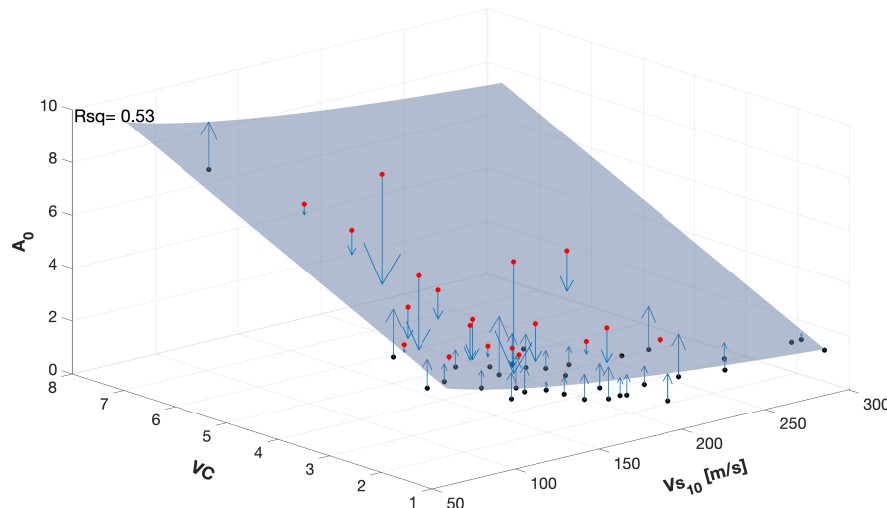

**Figure A2.** 3D-Plot of the two main parameters that define amplification; the velocity contrast and $Vs_{10}$. The pale blue surface depicts the fitting function between the parameters and divides the data points where the red points are above the surface and black points below. Blue arrows indicate the difference (error) between the surface and data point.

## Appendix B: Geological models

### B1    GeoTOP

GeoTOP schematizes the shallow subsurface of the Netherlands in voxels measuring 100 by 100 by 0.5 m (x ,y, z) up to a depth

of 50 m below ordnance datum (Stafleu et al., 2011, 2021). Each voxel contains estimates of the lithostratigraphic unit the voxel

belongs to and the lithologic class (including a sand grain-size class) that is representative for the voxel. GeoTOP is publicly



available from the web portal of TNO – Geological Survey of the Netherlands (GDN; https://www.dinoloket.nl/en/subsurface-models). GeoTOP is constructed using some 275,000 borehole descriptions from DINO, the national Dutch subsurface database operated by GDN (https://www.dinoloket.nl/en/subsurface-data), complemented with some 125,000 borehole logs from Utrecht
University in the central Rhine-Meuse river area. The modelling procedure involves four steps: First, the borehole descriptions are interpreted into standardized lithostratigraphic units with uniform sediment characteristics. Given the large number of boreholes, automated lithostratigraphic interpretation routines (Python scripts) were developed. These routines combine digital maps, stratigraphic rules (e.g. superposition) and lithologic criteria (e.g. main lithology, admixtures, grainsize and shell content, amongst other criteria) to determine the depth of the top and base of the lithostratigraphic units in each of the bore-
hole descriptions. Next, 2D interpolation techniques are used to construct surfaces bounding the bases of the lithostratigraphic units as observed in the boreholes. The interpolation algorithm allows for the calculation of a mean depth estimate of each surface and its standard deviation. Subsequently, all surfaces are stacked according to their stratigraphical position, resulting in a consistent layer-based model with estimates of top and base of each lithostratigraphic unit. Top surfaces are derived from the bases of the overlying units. The surfaces are then used to place each voxel in the model within the correct lithostratigraphic
unit. In the third step, the borehole descriptions are revisited and classified in six different lithologic classes ('peat', 'clay', 'clayey sand  sandy clay', 'fine sand', 'medium sand' and 'coarse sand and gravel'). In the last modelling step, a 3D stochastic simulation is performed for each lithostratigraphic unit separately. The simulation results in 100 equiprobable realizations of lithologic and grain-size class for each voxel. Post-processing of the realizations results in probabilities of occurrence as well as a 'most likely' estimate of lithologic and grain-size class. This 'most likely' estimate is used in the construction of the seismic
site-response zonation map (Appendix C).

## B2    NL3D

To date, the GeoTOP model covers about 70% of the country (including inland waters such as the Wadden Sea). For the missing areas we have used the lower-resolution voxel model NL3D, which is available for the entire country (Van der Meulen et al., 2013). NL3D models lithology and sand grain-size classes within the geological units of the layer-based subsurface
model DGM (Gunnink et al., 2013) in voxels measuring 250 by 250 by 1 m (x ,y, z) up to a depth of 50 m below ordnance datum. NL3D uses a much simpler modelling procedure than GeoTOP: First, the borehole descriptions are interpreted by intersecting each borehole with the top and base raster layers from the DGM model. The resulting stratigraphical interpretations are geometrically consistent with the DGM model, but not necessarily consistent with the borehole descriptions (e.g., a borehole interval describing 'sand' may erroneously fall within a unit that is characterized by clay deposits). Second, the surfaces of the
DGM model are used to place each voxel in the model within the correct lithostratigraphic unit. DGM is a layer-based model using a smaller dataset of some 26,500 manually interpreted borehole descriptions from the DINO database. Consequently, it is less refined than GeoTOP. For instance, DGM combines all Holocene formations in a single unit, whereas GeoTOP features some 25 different Holocene formations, members and beds. The third and fourth steps are identical to the ones described for GeoTOP. The resulting NL3D model has a similar 'most likely' estimate of lithologic and grain-size class which is used in the
construction of the seismic site-response zonation map (Appendix C).



## B3 Model uncertainty

The current version of GeoTOP covers about $28,605\,\mathrm{km^2}$ using some 400,000 boreholes. This implies that only about 7% of the voxels at land surface contain a borehole. Moreover, this number rapidly decreases with depth because many boreholes are quite shallow. Therefore, the lithostratigraphic unit and the lithologic class of almost all voxels are estimated on the basis of nearby borehole descriptions. As a 'rule-of'-thumb', the limited amount of data available deeper than $30\,\mathrm{m}$ below land surface strongly reduces the quality of the lithologic class estimates of GeoTOP (Stafleu et al., 2021). For NL3D, this number is $15\,\mathrm{m}$.

## B4 Applicability

GeoTOP and NL3D model the subsurface at a regional to subregional scale that is suitable for applications at the levels of province, municipality and district. The models are not suited for applications that require a finer scale at the level of streets or individual buildings.

## Appendix C: Workflow site-response map

The steps below describe the procedure used to assign the appropriate sediment (site-response) class to each of the voxel-stacks in GeoTOP and NL3D, as exemplified in Figure C1. A voxel-stack is the vertical sequence of voxels at a particular (x,y)-location in GeoTOP or NL3D. At each voxel there is an estimate of the lithostratigraphic unit and the lithologic class (Appendix B).

- **A.1** Calculate the cumulative thickness for each of the lithologic classes ('peat', 'clay', 'clayey sand and sandy clay' and 'sand') in the models for multiple depth intervals (5,10, 20 and $50\,\mathrm{m}$). The thicknesses of the lithologic classes 'fine sand', 'medium sand' and 'coarse sand  gravel' have been added together in the superclass 'sand'.

- **A.2** Calculate the depth of the top of the first consecutive sequence of sand with a minimum thickness of $1.5\,\mathrm{m}$ (GeoTOP) or $2\,\mathrm{m}$ (NL3D). This depth is further referred to as 'top sand'. In general, 'thick' sequences of sand represent the stiffer Pleistocene sediments. In other cases, they may represent Holocene sediments of, for example, the fluvial channel belt systems of the Rhine and Meuse, or the coastal dunes. These sands form the contrast with the overlying soft sediments ('peat', 'clay' and 'clayey sand and sandy clay'). Voxel-stacks containing a continuous Pleistocene clay sequence (Elsterian tunnel valleys) are included in the depth of the first sand (top sand), since no amplification is estimated here with the HVSR of site N02.

- **A.3** Calculate the percentage of each lithologic class above the 'top sand'. These percentages play an important role in assigning sediment site-response classes as described in steps **A.8** and **A.9**.

- **A.4** If anthropogenic deposits reach up to depths larger than $2\,\mathrm{m}$, no sediment class is assigned. Anthropogenic activities have modified the near-surface composition at many locations in the urbanized areas of the Netherlands. The lithologic class of these sediments is unknown. Therefore, we are not able to assign a sediment site-response class to those locations.





- **A.5** If bedrock outcrops or occurs at a depth smaller than 2 m, the site is assigned to Class I. The depth criterion is set at a maximum of 2 m since a deeper top bedrock would lead to a top layer with a possible resonance in the 1-10 Hz frequency band and hence a different site-response class. The top of the bedrock is determined from the DGM model (Gunnink et al., 2013) (top surfaces of the Houthem, Maastricht and Gulpen formations).

- **A.6** If bedrock in the eastern and southern part of the country occurs at a depth smaller than 100 m, the sediment site-response class is set to V. These are sites where the layer on top of the bedrock could yield a resonance in the 1-10 Hz band, which resonance has not sufficiently be calibrated to assign an $AF_{NL}$. Class V thus corresponds to sites with a currently unknown amplification potential. The top of the bedrock is determined from the DGM-deep model (Gunnink et al., 2013) (top surfaces of the Rijnland and Chalk groups).

- **A.7** If 'top sand' is less than 2 m, the site-response class is set at II. Examples of HVSR curves with 'top sand' less than 2 m, do not exhibit any peak amplitude due to the absence of a resonating soft layer on top of a stiffer one.

- **A.8** If 'top sand' is between 2 and 3 m, the lithologic distribution of the overlying soft sediments determine if the sediment site-response class will be II, II or IV. Examples of HVSR-curves with 'top sand' between 2 and 3 m show peaks for certain lithological successions, forming a resonating layer. Class II is assigned if the overlying sediments contain more than 60% sand. Class III is assigned if the overlying sediments are mainly composed of clayey sand  sandy clay; and class IV if clay and peat dominate. We do not elaborate on the exact percentages tot decide between Class III and IV. While testing the different criteria, this step appeared to be quite sensitive, and needed the implementation of several exceptions to the general rule.


- **A.9** If 'top sand' is larger than 3 m, the approach is basically the same as in **A.8**. Class II is assigned if the overlying
sediments contain more that 60% sand. However, the exact criteria to decide between class III and IV differ from those in **A.8**.

*Code and data availability.*  We provide we provide three (for each geological model used) ArcAscii-files containing the coordinates and the corresponding classification to be used for plotting the site-response zonation map. A Matlab code is provided for reading and plotting the data-files. Additionally, a high resolution version of the site-response zonation map (Figure 12) is provided.

*Author contributions.*  Janneke van Ginkel: method development, data analysis, construction of the map and underlying criteria, writing of manuscript with input from all co-authors. Elmer Ruigrok: daily advisor, input on data analysis method and results, performed text input. Jan Stafleu: provided the GeoTOP and NL3D data, supported and quality checked the voxel-stack analysis and advised on the criteria for the construction of the site-response zonation map and performed text input. Rien Herber: promotor, initiator of this research project, advisor, method development, final text editor.


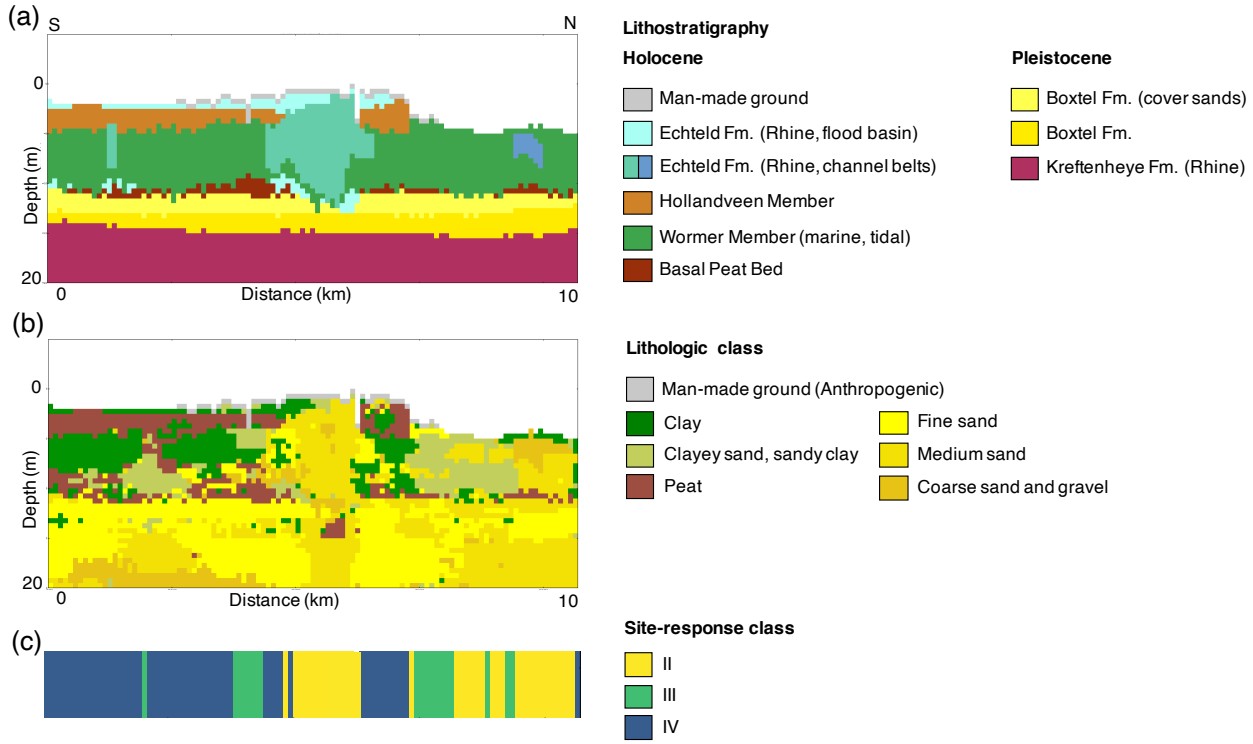

**Figure C1.** Cross-section through the GeoTOP voxel model: a) lithostratigraphy, b) lithologic class and c) corresponding sediment site-response class. The cross-section runs S-N through the city of Alphen aan den Rijn, situated on a sandy Holocene channel belt of the river Oude Rijn ('Old Rhine'). For location see Figure 13b). Class III and IV appear where soft, Holocene sediments (clay and peat) are overlying stiff Pleistocene deposits (sand). However, where Holocene sediments are sandy, such as in the channel belt in the center of the cross-section, Class II occurs.

*Competing interests.* The authors declare no competing interests.

*Acknowledgements.* This work is funded by EPI Kenniscentrum. Ambient noise and earthquake recordings were provided by KNMI and are publicly available through the website (http://rdsa.knmi.nl/dataportal). Figures are produced in Matlab. We would like to thank Deltares for the use of the SCPT data and lithological interpretations and TNO for the use of the 3D geological models and maps.



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
