# Peer review of "Development of a country-wide seismic site-response zonation map for the Netherlands"

_Natural Hazards and Earth System Sciences, 2021_

## Referee Comment (RC1)

Review for the article titled as:

"Development of a country-wide seismic site-response zonation map for the Netherlands" by van Ginkel *et. al*.

This interesting paper attempts to assess site response for the Netherlands for a first identification of regions with increased seismic hazard potential. Ambient noise HVSR amplitude (A0), amplification factors (AF) and empirical transfer functions (ETF) were retrieved by means of data recorded by the seismic network (consisting of borehole and surface seismometers) of the Royal Netherlands Meteorological Institute (KNMI, 1993) across the Netherlands. S-wave velocity profile from a Seismic Cone Penetration Test (SCPT) available for some boreholes were also used together with detailed 3D geological subsurface models GeoTOP and NL3D. All these data were used to derive empirical relationships between measured amplification in the time and frequency domain, estimated amplification from the ambient noise field and the local lithostratigraphic conditions and a zonation map as well.

The results of this work is based on extensive geophysical and some seismological measurements. Though these techniques are widely used to address site effects, the attempt to provide a 3D model of the site through the obtained results is, to a certain degree, a novel approach and definitely adds extra value. This reviewer recognizes that the paper contains interesting results. The content of the article is also very relevant and in line with the focus of the special issue. The paper is generally well organized and well written. However, there is still scope for enhancing the writing quality. For example, the authors should avoid some repetitions.

The scope fits the subject "Natural Hazards and Earth System Sciences". So, this reviewer believes that the paper should be considered for publishing and recommend minor revisions.

GENERAL COMMENTS & QUESTIONS

Line 171: At this depth, 95% of the Dutch subsurface is composed of these sediments

Line 191: In this study we compute amplification factors (AF) in the time domain from the G-network earthquake recordings. We compute the AF for each borehole site by taking the ratio of the maximum amplitudes recorded at the surface and the 200 m deep seismometer.

Line 194: The amplitude at the surface was divided by a factor of 2 order to remove the effect of free surface amplification.

Line 196 – 199: We decided to adapt the frequency band and seismometer depth to obtain an AF that is more representative for use on a national scale than the AF used in the region of Groningen. Hence in this paper, the AF is calculated between the seismometers at surface and at 200 m depth, for a frequency band of 1-10 Hz. The AF is determined in the time domain and therewith it provides an average amplification over the applied frequency band.

Line 210: In this study we compute the ETF between the radial component of the seismometers at surface and at 200 m depth (ET F200).

Line 215: demonstrating that most amplification develops in the top 50 m of the sediment cover which is supported by

**Commented [SU1]:** We decided to adapt the frequency band and seismometer depth to obtain an AF that is more representative for use on a national scale than the AF used in the region of Groningen. From the G-network earthquake recordings, we compute the AF for each borehole site by taking the ratio of the maximum amplitudes recorded at the surface and the 200 m deep seismometer.

**Commented [SU2]:** Why a value of 2?

**Commented [SU3]:** in order

**Commented [SU4]:** The AF is calculated between the seismometers at surface and those at 200 m depth, for a frequency band of 1-10 Hz. It is determined in the time domain and therewith provides an average amplification over the applied frequency band.

**Commented [SU5]:** Delete "In this study"

**Commented [SU6]:** are developed

Line 245 - 246: Whereas the ETF peak amplitudes represent maximum amplification (at peak frequencies which vary from site to site), the empirical relationship between the HVSR A0 and AF is of most importance for the construction of the site-response map.

**Commented [SU7]:** Why?

Line 250: However, recent studies (Castellaro et al., 2008; Kokusho and Sato, 2008; Lee and Trifunac, 2010) have drawn attention to the fact that using only V s30 as proxy for site-response is inadequate,

**Commented [SU8]:** However, studies from Castellaro et al. (2008); Kokusho and Sato (2008); Lee and Trifunac (2010) have drawn attention. Delete "recent" because these studies are older than those cited in the previous sentence.

Line 253: Hence, the shear-wave velocity ratio between the top and base layer is introduced as a proxy for site amplification by Joyner and Boore (1981) and further explored by Boore (2003)

**Commented [SU9]:** Delete "Hence" for the same reason as previously

Line 255: the AF is fitted, using A0 = x1 +x2e x3$^{Vs}$

**Commented [SU10]:** What are x1, x2, x3?

Line 260: Secondly, from the SCPT data we derive the depth and size of the velocity contrast (VC) by dividing the shear-wave velocity values for each 1 m interval by the maximum value over the full 30 m is taken as the VC-value.

**Commented [SU11]:** Something is missing

Line 265: On the other hand, the correlation between the AF and the VC is less, meaning this parameter is inferior to the AF.

**Commented [SU12]:** Do you mean that the influence of this parameter is the least?

Line 267: A large VC-value is leading to resonance in the near-surface, which is expressed in high amplitude peaks of the HVSR

**Commented [SU13]:** leads

Line 287: For the TERZ borehole the ETF is displaying similar curve characteristics as the HVSR estimations.

**Commented [SU14]:** displays

Line 291: The borehole ETFs confirm that most of the amplification develops in the top 50 m (Figure 5) of the sedimentary cover

**Commented [SU15]:** are developed

Line 292: The top 10 m (Figure 7)is

**Commented [SU16]:** (Figure 7) is

Line 306: The main reason of is that we

**Commented [SU17]:** Missing a word

Line 311: sites with bedrock at depths shallower than 100 m fall into Class V. For Class V, the resonance over the complete unconsolidated

**Commented [SU18]:** Avoid repetition. For example, "than 100 m fall into Class V for which the resonance over the complete unconsolidated"

Line 356: As a result, we present the national site-response zonation map (Figure 12), were each class characterises a certain level

**Commented [SU19]:** where

Line 361: Typically, at these places there is large model uncertainty. For example in north-east Noord-Holland

**Commented [SU20]:** uncertainty, for example

Line 367: the HVSR of the seismometers (e.g. ALK2, Figure 8) deficit any peak due to the absence of an velocity contrast

**Commented [SU21]:** a

Line 363: The geological model at these locations presents large portions of clayish sand, resulting in class category III, while the HVSR curves exhibit distinctive, high amplitude peaks, demonstrating local conditions related to class IV.

**Commented [SU22]:** According to you, what can explain such a discrepancy? What show for example the corresponding ETF?

Line 370: Most of the northern part of this region is Class II due Pleistocene sands

**Commented [SU23]:** due to Pleistocene

Line 394: the site-response map (Figure 12) exhibits an which is rather similar to the geological map (Figure 1).

**Commented [SU24]:** A word is missing

---

## Referee Comment (RC2)

This document presents a review of the manuscript titled as "Development of a country-wide seismic site-response zonation map for the Netherlands" by van Ginkel et al.

As the title indicates, the research work involves the development of a new seismic zonation map for the Netherlands. The authors attempted to combine geological, geophysical, and seismological data in order to estimate and interpret site effects. This reviewer recognizes that this work is based on the analysis of a very large dataset; hence, it involves an important volume of work. Such a work could definitely contribute to the understanding of site-effects induced by shallow subsurface geology. The zonation map could also be considered as a novel contribution in the seismic hazard assessment of the Netherlands. However, it's a pity that this work has been very poorly presented throughout the article.

Firstly, the use of English language is very raw. The entire article is full of long and complex sentences, expressing multiple ideas at the same time in wrong grammatical structures. The underuse, and sometimes misuse of punctuations (e.g. comma) makes the sentences even more incomprehensible. Such a long article with tedious and incorrect English sentences can easily distract the readers. The mistakes are too many to point out. Some examples have been given below. This reviewer strongly recommends that this article must be reviewed by a proficient English user.

Secondly, the organization of the article is not up to the standards. The writing needs to be more coherent within each segment, and should present a concise and unambiguous idea. For example, the introduction is composed of generalized statements and often unrelated specific information. This section should rather state the motivation of the work and prepare readers for the structure of the article. Generally speaking, it should provide a context first, then show the need of the work, then indicate what have been done in order to address the need, and finally preview the structure of the rest of the article. The principal elements are more or less there in the current article but they are presented in an unorganized manner.

Finally, the article still lacks in scientific soundness. The methods are not outlined clearly and the calculations are not shown with proper mathematical formulations. It's not evident which sites or earthquakes have been used for which calculation. Mainly the final summary of the results are presented rather than showing the intermediate steps. The interpretations of the results are ambiguous and incomplete. Many hypotheses and assumptions have been made without any justification. Therefore, this reviewer is doubtful about the traceability of the results.

Considering all these issues, this reviewer believes that this article is not eligible for publication at its current state. However, the authors are encouraged to resubmit it after a rigorous modification and amelioration of its editorial and scientific quality.

The main remarks are elaborated below (L= Line):

**Main Scientific Questions:**

- L165: What is the justification for setting the reference bedrock at 200 m depth? This hypothesis has been supported neither by the geological profile nor by geophysical measurement. Is the geology at 200 m depth same everywhere? Is there any shear-wave velocity profile that shows that the formation at 200 m depth can be characterized as rock?

- The authors mention that "*This depth and corresponding average shear-wave velocity forms the basis from which the site-response and corresponding amplification factors (AFs) are estimated in the next sections.*" Where is this shear-wave velocity defined? If the bedrock is defined without any justification, the estimation of amplification in the entire work becomes highly questionable. The authors did mention something about Groningen network. However, they do not show the location of this network with respect to their data. It's also very unclear how the bedrock has been identified from the G-network. How can a Vs 500 m/s be characterized as rock condition? How can this be applicable to all sites, especially in the south where there are older formations?

- Figure 1: What does the geological map correspond to? Does it show the geology at the surface or at any specific depth? The figure does not show geographical coordinates.

- The description of the geology seems a bit incomplete. Even though this work uses two 3D geological models, there is not enough discussion about the geology at depths (e.g., at the base of the Quaternary or below).

- Figure 2: Lat/Lon should be shown at least for two points on both axes.

- L 190: The AFs are calculated from the G-network but the location of this network has not been shown. Do the boreholes with SCPT belong to this network?

- The calculation of AFs need to be elaborated in mathematical terms and the signal processing aspects need to be explained better. The calculation of AFs for the event shown in Figure 3 can be presented as an example. The 1D geology and Vs profile at that location could also be presented to show if the AF could be explained/interpreted.

- Which M>2 earthquakes have been used for the AF calculation – the induced or tectonic ones? How many earthquakes are there? What are their magnitude-distance distributions? Have they been selected based on good signal-to-noise ratio?

- Why are the AFs calculated in such large frequency bands? Such results provide very little resolution for the interpretation of the amplification. Is there an estimate of the Vs of the sedimentary layer? Is it possible to verify if the fundamental resonance of amplification is captured within 1-5 Hz band?

- The authors mention that the high AFs in 1-5Hz band is due to fundamental resonances but they do not provide any evidence to support that.

- L205: Once again, which earthquakes have been used to compute the ETFs? The computation of the ETFs need to be elaborated with appropriate examples. This reviewer is not convinced by the interpretation of the ETFs. Do the ETF50 and ETF200 have similar amplitudes at all sites? Can it be supported by the geology of some example sites?

- Figure 5: The visibility of this Figure is poor. It's difficult to verify the comparison among the curves. The X-axis is not graduated at all.
- L235: Which are the sites where HVSR, ETF and AF all are measured? Please show on the map. How many earthquakes (and their M-R distribution) are available for those sites? Figure 7 is not well explained and the Figure title is also unclear.What are the values plotted there? At which frequency?
- L 255: It seems that the Vs10, Vs20 and Vs30 values are taken from one set of sites and the Vs50 is taken from another. Is that so? Which of these sites correspond to the ones where AFs and ETFs are estimated? How far away the other sites are?
- L 259: How are the depth and size of velocity contrast derived? It's not very clear from the description. Please provide mathematical formulation.
- What is the rational of using the particular functional forms for fitting AF with Vs and VC? What are x1, x2, x3?
- L 280: Where are these stations located? They could not be found anywhere in the article?
- L 292: It's not evident to this reviewer that the borehole ETFs show most amplification within 50 m depth. Only one random example has been shown in Figure 5. The 50m, 200m depth values seem more like mere assumptions of the authors. In Figure 7, the fit between AF and Vs seems more or less similar for Vs10, Vs20, V30, Vs50. It rather seems that the functional relation could be slightly different in case of Vs10, Vs20 compared to Vs30, Vs50. As none of these results have been explained/supported throughout the paper by concrete geological and geophysical information, the summery and interpretation of the results seem very ambiguous.
- This reviewer is also doubtful about the site classification approach in this article and, hence, about the entire zonation. Replicating the HVSR-AF correlation obtained from the limited Groningen area for the entire country seems a bit heavy-handed. In other studies (e.g., Perron et al.), HVSR has been shown as a complementary parameter for amplification prediction within an area where already some estimates of AF exist. The site-specific nature of amplification must be addressed in a zonation approach.
- This reviewer suggests the authors to highlight the geology more and verify/interpret/constrain the results in terms of the geology of the measurement sites. It's important to explain the effects of the subsurface structure/geology on the amplification rather than drawing purely statistical functional correlations.

**Editorial Remarks (some examples from numerous mistakes):**

- L5: *"The shallow geology of the Netherlands consists of a very heterogeneous soft sediment cover, which has a strong effect on seismic wave propagation,  in particular on the amplitude of ground shaking, resulting in significant damage on structures ."* **The phrase with 'despite' makes the sentence confusing. The use of 'events' is also confusing.**

- L10: "*For this, we combine ambient vibration and earthquake recordings using resp. the horizontal-to-vertical spectral ratio method (HVSR), borehole empirical transfer functions (ETFs) and amplification factors (AFs)."* **How is it possible to combine noise and earthquake recordings?**

  "*This enables us to define an empirical relationship between*  *the amplification estimated from earthquakes by using the ETF and the AF, and that*  *estimated from ambient vibration*  *by using the HVSR* *. Therewith, we show that the HVSR can be used as a first proxy for amplification."* **Grammatically incoherent sentence. The use of 'therewith' is not the best choice.**

- L15: "*The resulting peak amplitudes largely coincide with the in-situ lithostratigraphic sequences and the presence of a strong velocity contrast in the near-surface.*" **Very confusing sentence. Do the authors means that 'the resonance frequencies of peak amplifications can be explained by the velocity contrasts in the lithostratigraphy'?**

- L 25: "*Site conditions may be retrieved from available global datasets and the ground-shaking estimation is based on ground-motion prediction equations.*" **The point of this sentence is unclear.**

  "*Site-response estimation requires detailed geological and geotechnical information of the subsurface, which can be retrieved from in-situ investigations, however, this is a costly procedure.*" **For the sake of clarity, the sentence needs to be split into two.**

- L47: "*Hence multiple studies were performed on ground-motion modeling including the site amplification factor for the Groningen region, which forms an excellent study area due to the permanently operating borehole seismic network.*" **This sentence makes little to no sense.**

- L115: "*The Netherlands experiences two types of seismicity; firstly, earthquakes in the south-east are caused by deep tectonic processes and secondly, induced seismicity at shallow depths triggered by exploitation of gas fields.*" **Grammatically incoherent.**

- L121: "*They observed great variety in ground-motion amplitudes over different stations which is very likely a site effect of shallow sedimentary deposits."* **Makes no sense**

- L155: "*The extensive data set recorded with the Groningen borehole network provides the opportunity to derive empirical relationships between measured amplification in the time and frequency domain, estimated amplification from the ambient noise field and the local lithostratigraphic conditions."* **Makes very little sense**

---

## Author Comment (AC1)

Dear Reviewer,

Thank you for your positive remarks and insightful comments on the paper. We appreciate the time and effort that you have dedicated to our manuscript. We have discussed your main technical suggestions and summarized the outcome below. The small editorial suggestions will be also incorporated in the revised manuscript, which will be uploaded in a later stage.

- Line 191-199: *Changed the sentences in this section according to your suggestions.*
- Line 194: Why a value of 2?
  *We use a factor of 2 because at the surface, the up- and down-going waves are recorded at the same time. By the division by 2, the amplitude of the upgoing wave is retrieved. We will add a sentence in the updated manuscript.*
- Line 245-246: Why is the relationship between HVSR and AF of most importance?
  *The relationship between the HVSR $A_0$ and the AF is used to obtain an AF per class for the zonation map (Section 6.3). HVSR records are available throughout the country while the AF is not. These lines are rewritten for more clarity.*
- Line 255: What are x1, x2 and x3?
  *You have raised a good point here and accordingly in the new manuscript we changed the x1, x2, x3 to a,b,c to circumvent confusion with Cartesian coordinates. a,b and c are the three unknown coefficients to be fitted. This line is rephrased for more clarity.*
- Line 265: Do you mean that the influence of this parameter is the least?
  *Indeed, we mean the influence is less. But removed the sentence since it was in repetition with the next sentence.*
- Line 363: According to you, what can explain such a discrepancy? What show for example the corresponding ETF?
  *Thank you for pointing this out. Unfortunately, for this area we cannot compute an ETF since it is seismically quiet. With this example we like to point out that the GeoTOP model is a model, and interpolated between the data points. This adds extra uncertainty to the map, which is discussed in Section 6.3.*
- Line 394: A word is missing.
  *Thanks, yes here a word is missing. Added 'regional pattern'.*

We hope we cover your comments and are willing to respond to any further questions and suggestions you may have.

Sincerely,

Janneke van Ginkel, Elmer Ruigrok, Jan Stafleu and Rien Herber

---

## Author Comment (AC2)

Dear Reviewer,

Thank you for your comments on the paper. We appreciate the time and effort that you have dedicated to our manuscript. We have carefully reviewed the manuscript and rephrased some sections to better communicate the main message. In the introduction, the structure of the article is more emphasized. The sites and earthquakes used for the calculations are presented in Figure 2. For interpretation and justification of the results, we included more details on the methods and the effect of the subsurface geology on amplification.

We have discussed your main scientific questions and summarized the outcome below. The editorial suggestions will be also incorporated in the revised manuscript, which will be uploaded in a later stage. We believe that with the incorporation of your suggestions, the manuscript has improved.

**Response to main scientific questions:**

- L165: What is the justification for setting the reference bedrock at 200 m depth? This hypothesis has been supported neither by the geological profile nor by geophysical measurement. Is the geology at 200 m depth same everywhere? Is there any shear-wave velocity profile that shows that the formation at 200 m depth can be characterized as rock?

  *Thanks for pointing this out and it is a relevant thought. Indeed, in the Netherlands at 200 m depth there is no real solid rock. Generally, amplification is determined with respect to a reference bedrock, but such reference site does not exist in the Netherlands. Studies by e.g., Poggi et al. (2011) also derive a reference site that does not correspond to an actual bedrock site. It is defined as an (average) S-wave velocity profile. We further simplify the approach by taking the elastic conditions at 200 m depth in Groningen as a reference, from which we define amplification. Similar conditions can be found (at the same or other depths) in most of the Netherlands.*

  *We define reference conditions at depth with a shear-wave velocity of 500 m/s. These are the in-situ shear-wave velocity values that are found, on average, at 200 m depth, based on studies on the Groningen borehole network from Hofman et al. (2017) and Kruiver et al. (2017). Overall, the subsurface composition throughout the country is quite uniform at 200 m depth and consists of semi-consolidated clastics. The (relatively few) locations with deviating subsurface conditions are evaluated separately and clustered in class V. You raised a good point that the term 'rock' in line 169 is misleading here. Section 4.1 is rephrased for more clarity.*

- The authors mention that "This depth and corresponding average shear-wave velocity forms the basis from which the site-response and corresponding amplification factors (AFs) are estimated in the next sections." Where is this shear-wave velocity defined? If the bedrock is defined without any justification, the estimation of amplification in the entire work becomes highly questionable. The authors did mention something about Groningen network.

However, they do not show the location of this network with respect to their data. It's also very unclear how the bedrock has been identified from the G-network. How can a Vs 500 m/s be characterized as rock condition? How can this be applicable to all sites, especially in the south where there are older formations?

*Some of these questions are already answered in the previous comment about how we define the reference shear-wave velocity, using conditions as found at 200 m depth at stations of n the Groningen borehole network. The location of the Groningen network is shown in the updated version of Figure 2 (see figure next page).*

*Overall, the subsurface composition throughout the country is quite uniform at 200 m depth. At some locations similar conditions can be found at shallower depths. The locations where the reference subsurface conditions do not occur in the top few hundred meters are evaluated separately and clustered in Class V. For example in the very south of the Netherlands, Class V can be found.*

- Figure 1: What does the geological map correspond to? Does it show the geology at the surface or at any specific depth? The figure does not show geographical coordinates.

  *It is the geological map for the surface geology, we added this remark to the caption. Also, the coordinates have been added.*

- The description of the geology seems a bit incomplete. Even though this work uses two 3D geological models, there is not enough discussion about the geology at depths (e.g., at the base of the Quaternary or below).

  *Thanks for pointing this out. In Section 2, additional details of the geology at larger depths are provided.*

- Figure 2: Lat/Lon should be shown at least for two points on both axes.

  *Indeed, thanks. More lat/long coordinates are added to the figure.*

- L 190: The AFs are calculated from the G-network but the location of this network has not been shown. Do the boreholes with SCPT belong to this network?

  *For more clarity, we added a subfigure to Figure 2 showing the Groningen borehole network and the locations of the local earthquakes used for AF and ETF computations. The purple triangles are the locations with SCPTs available.*

  *Find below the updated figure and caption:*

[Figure]

*Figure 2: Map of the Netherlands depicting epicentres of all induced (Mw 0.5-3.6, orange) and tectonic (Mw 0.5-5.8, yellow) earthquakes from 1910-2020. The diameter of the circles indicates the relative earthquake magnitude. The triangles represent the surface location of the borehole stations (blue), borehole stations with SCPT measurement (purple) and single surface seismometers (green). The triangles with red outlines depict the locations of example HVSR curves presented in Figure 9. The inset in the north-east depicts the location of the Groningen borehole network (G-network). The 19 (Mw ≥2) induced earthquakes in this panel are used for the AF and ETF computations. Coordinates are shown within the Dutch National Triangulation Grid (Rijksdriehoekstelsel or RD) and lat/long coordinates are added in the corners for international referencing.*

- The calculation of AFs need to be elaborated in mathematical terms and the signal processing aspects need to be explained better. The calculation of AFs for the event shown in Figure 3 can be presented as an example. The 1D geology and Vs profile at that location could also be presented to show if the AF could be explained/interpreted.

  *It is a good suggestion to add the G24 velocity profile and corresponding lithology. Figure 3 is updated accordingly. The calculations of AFs are also explained in more detail in the updated manuscript. We decided not to show the equation since the procedure is easily conveyed in words.*

- Which M>2 earthquakes have been used for the AF calculation – the induced or tectonic ones? How many earthquakes are there? What are their magnitude-distance distributions? Have they been selected based on good signal-to-noise ratio?

*Thanks for pointing this out since this is important information to add. We used 19 M$\geq$ 2 local induced earthquakes, added this number to the text. In Groningen, all earthquakes are induced (see Figure 2). We deliberately used the M$\geq$2 earthquakes since lower magnitudes are not recorded across the entire network. By using 19 earthquakes with different magnitudes and distances, the magnitude-distance relationship (if present) is averaged out. Furthermore, we also included a signal-to-noise threshold to obtain reliable results.*

- Why are the AFs calculated in such large frequency bands? Such results provide very little resolution for the interpretation of the amplification. Is there an estimate of the Vs of the sedimentary layer? Is it possible to verify if the fundamental resonance of amplification is captured within 1-5 Hz band?

*You have raised a good point here. The 1-10 Hz band captures the full range of possible resonance frequencies of most structures in the Netherlands and is the interval of interest for earthquake engineering purposes. By division into multiple spectra ordinates, we should design also multiple site-response zonation maps. This detailed revision is a good suggestion for future work.*

*Vs30 profiles are available from the SCPT's obtained next to the Groningen boreholes. From the relationship between Vs, and sediment thickness of the Holocene infill (f0=Vs/4\*h) we can conclude that the resonance frequencies are mostly in the band of 1-5Hz. Van Ginkel et al. (2019) explains the relationship between the amplification and Holocene infill.*

- The authors mention that the high AFs in 1-5Hz band is due to fundamental resonances but they do not provide any evidence to support that.

*We refer to the paper Van Ginkel et al. (2019) to show that the fundamental resonances occur in this band.*

- L205: Once again, which earthquakes have been used to compute the ETFs? The computation of the ETFs need to be elaborated with appropriate examples. This reviewer is not convinced by the interpretation of the ETFs. Do the ETF50 and ETF200 have similar amplitudes at all sites? Can it be supported by the geology of some example sites?

*The updated manuscript contains more detailed information on the earthquakes used. Additionally, a reference is added to the methodology of calculating transfer functions. Almost the entire Netherlands is covered with thick (> 200m) sediments. At 200 m the subsurface contains Pleistocene clastic sediments and only the top tens of meters comprise the (very) soft unconsolidated sediments. So almost everywhere the ETF200 resembles the ETF50. In Groningen, we observe at all sites with a computed ETF a relationship with the geology. It is a good point you raised; hence the updated manuscript includes a few lines on the relationship with the geology*

- Figure 5: The visibility of this Figure is poor. It's difficult to verify the comparison among the curves. The X-axis is not graduated at all.

*You have raised a good point here and accordingly the figure will be updated with larger fonts and a better distinction among the curves.*

- L235: Which are the sites where HVSR, ETF and AF all are measured? Please show on the map. How many earthquakes (and their M-R distribution) are available for those sites? Figure 7 is not well explained and the Figure title is also unclear.What are the values plotted there? At which frequency?

  *For the ETF and AF, we use 19 local earthquakes. These are all the $M \geq 2$ earthquakes recorded with the Groningen borehole network since it was deployed in 2015. In the new manuscript we will add a figure in Figure 2 (shown above) illustrating the earthquake epicentres and the boreholes used for the computations.*

  *The updated manuscript will contain an improved description and caption of Figure 7. In Figure 7, the y-axis are the AF values computed for the Groningen borehole network for 1-10 Hz, as presented in Section 4.2. Furthermore, subfigure labels are added to each panel in order to make references in the text. Additionally, the description of the axis labels is adjusted for more clarity.*

- L 255: It seems that the Vs10, Vs20 and Vs30 values are taken from one set of sites and the Vs50 is taken from another. Is that so? Which of these sites correspond to the ones where AFs and ETFs are estimated? How far away the other sites are?

  *The Vs10, Vs20 and Vs30 are values from the SCPT data obtained adjacent to 53 of the 68 borehole sites. Figure 2 displays the borehole locations where SCPTs have been taken. The Vs50 is computed based on records at the surface and 50 m deep seismometer for each borehole, as presented in Hofman et al., (2017), comprising the same sites as the SCPTs. So, all velocity-values used are at the borehole sites from which also the ETF, HVSR and AFs are computed.*

- L 259: How are the depth and size of velocity contrast derived? It's not very clear from the description. Please provide mathematical formulation.

  *Thanks for pointing out that this was missing. The velocity contrast is derived from the SCPT shear-wave velocity values. The contrast is computed by the division of the two different velocity values bounding each 1 m interval. This division is done for each 1 m interval over 30 m of SCPT records. The largest division value is defined as the velocity contrast (VC) and corresponding depth is the depth of the contrast (zVC).*

- What is the rational of using the particular functional forms for fitting AF with Vs and VC? What are x1, x2, x3?

  *Thanks for pointing this out, indeed this paragraph requires more clarity, accordingly in the new manuscript we change the x1, x2, x3 to a, b, c to avoid confusion with Cartesian coordinates. a, b and c are the three unknown*

*coefficients to be fitted. This line is rephrased for more clarity. The AF-Vs relationship exhibits an exponential fit, while the AF-VC is logarithmic.*

- L 280: Where are these stations located? They could not be found anywhere in the article?

  *Figure 2 is updated with the locations of the example HVSR (triangles with red outline)*

- L 292: It's not evident to this reviewer that the borehole ETFs show most amplification within 50 m depth. Only one random example has been shown in Figure 5. The 50m, 200m depth values seem more like mere assumptions of the authors. In Figure 7, the fit between AF and Vs seems more or less similar for Vs10, Vs20, V30, Vs50. It rather seems that the functional relation could be slightly different in case of Vs10, Vs20 compared to Vs30, Vs50. As none of these results have been explained/supported throughout the paper by concrete geological and geophysical information, the summery and interpretation of the results seem very ambiguous.

  *It is an important point you are raising here and accordingly we provide here an extra figure explaining the assumption that amplification largely occurs within the top 50m. Hence, Figure 3 below shows the ETF peak amplitudes for the ETF200, the ETF50 and the absolute difference (ETF200-ETF50) for the boreholes of the Groningen network. Overall, the small value for difference indicates that most amplification occurs in the top 50 m. Since the AF exhibits similar observations as discussed in van Ginkel et al. (2019) we decided not to include this in this manuscript.*

[Figure]

*Figure 1: Bar plot illustrating the absolute difference (yellow) between the peak amplitude over a frequency band of 1-10 Hz ETF200 (blue) and the ETF50 (red) for each borehole in the Groningen network.. ETF*

  *Secondly, you raised a good point here about missing the link between the Vs10-Vs20 fit and the geology. Accordingly, we added a few lines in section 4.5 about the relationship:*

*"In Groningen, the low-velocity and unconsolidated Holocene sediments have a thickness of 1-25 m and below these depths the velocities increase in the more compacted Pleistocene sediments. The reduced fitting quality of the Vs30 and Vs50 arises since the amplification develops mainly in the Holocene sediments (van Ginkel et al., 2019)."*

*Furthermore, Figure 8 is updated to show the link between the HVSR and corresponding sediment profile in order to add more information on the relationship between the seismological observation and the geology.*

[Figure]

*Figure 8: Each panel depicts a probability density function from ambient noise HVSR curves and sediment profile (Section 5) for 16 stations of the NL-network. The black line represents the mean HVSR and the red line in the panel of T06, T010 and TERZ represents the ETF calculated from 10 local earthquakes. The color bar in the lower right displays the HVSR probability range that is valid for all panels.*

- This reviewer is also doubtful about the site classification approach in this article and, hence, about the entire zonation. Replicating the HVSR-AF correlation obtained from the limited Groningen area for the entire country seems a bit heavy-handed. In other studies (e.g., Perron et al.), HVSR has been shown as a complementary parameter for amplification prediction within an area where already some estimates of AF exist. The site-specific nature of amplification must be addressed in a zonation approach.

*Thanks for pointing this out and as suggested in your review, we added more details of the methods and link to geology/geophysics to the updated manuscript. We believe the site classification approach presented is reasonable due to the following reasons:*

1. *In the Netherlands, the shallow geology is laterally quite uniform in terms of recent depositional history (see geological map, Figure 1). Therefore, the correlations obtained from the Groningen area are quite representative for the majority of the country. This area contains the most recent Holocene unconsolidated sediments, as well as the Pleistocene sand that cover a large part of the country. Hence, a large part of the typical NL sites is covered within the Groningen region.*
2. *For locations with a deviating Holocene or Pleistocene top 50 m, like the coastal dunes, we analyze the HVSR curves estimated from ambient vibrations recorded at seismometers throughout the country and link it to the geology at that site. This is to expand the sediment classification in order to address the site-specific nature of amplification.*
3. *We observe strong similarities in terms of curve characteristics and subsurface geology between the HVSR estimations throughout the country and the HVSR characteristics in Groningen. This confirms that amplification in the top 50 m sediment layer develops consistently throughout the country.*
4. *Each HVSR curve is linked to the sediment profile classification. This is justified with more details in Figure 4 of this response.*
5. *Due to the uniform sedimentation of the clastics, the AF-HVSR correlation derived from Groningen seems a reasonable first approach for country-wide AFs. With more data available on AFs at multiple locations, the AF-HVSR relationship can be optimized by using the approach described in e.g. Cultrera et al. (2014), Perron et al. (2018) and Panzera et al. (2021).*
6. *In order to correct for the locations with deviating and shallow 'bedrock' (<100 m) conditions, we designed class V. This setting is absent in Groningen so we are not able to define AFs here. This class needs further investigation for site-response and AF estimations and this is discussed in the manuscript.*
7. *We are aware that our approach comes with uncertainties and we address the AF uncertainty in Section 6.3.*
8. *Lastly, this manuscript presents a first approach for the zonation, when more data becomes available, the map can be updated accordingly. But for a first site-response estimation at locations with limited data available, we believe our approach is reasonable.*

- This reviewer suggests the authors to highlight the geology more and verify/interpret/constrain the results in terms of the geology of the measurement sites. It's important to explain the effects of the subsurface structure/geology on the amplification rather than drawing purely statistical functional correlations.

*Thanks for pointing this out and accordingly we have added more details on the geology and the relationship with the seismological and geophysical observations (see the previous points). Also, we refer to van Ginkel et al. (2019), which reference describes in detail the relationship between geology and amplification in Groningen.*

Your editorial remarks will be incorporated in the new manuscript. Furthermore, the manuscript is carefully read to address writing imperfections. We hope we cover your comments and are willing to respond to any further questions and suggestions you may have.

Sincerely,

Janneke van Ginkel, Elmer Ruigrok, Jan Stafleu and Rien Herber

**References**

Cultrera, G., De Rubeis, V., Theodoulidis, N., Cadet, H., & Bard, P.-Y. Statistical correlation of earthquake and ambient noise spectral ratios.Bulletin of earthquake engineering,12(4), 1493–1514, 2014.

Hofman, L., Ruigrok, E., Dost, B., and Paulssen, H.: A shallow seismic velocity model for the Groningen area in the Netherlands, Journal ofGeophysical Research: Solid Earth, 122, 8035–8050, 2017.

Kruiver, P. P., van Dedem, E., Romijn, R., de Lange, G., Korff, M., Stafleu, J., Gunnink, J. L., Rodriguez-Marek, A., Bommer, J. J., van Elk,J., et al.: An integrated shear-wave velocity model for the Groningen gas field, the Netherlands, Bulletin of Earthquake Engineering, pp.1–26, doi: 10.1007/s10518-017-0105-y, 2017a

Panzera, F., Bergamo, P., and Fäh, D.: Canonical Correlation Analysis Based on Site-Response Proxies to Predict Site-Specific AmplificationFunctions in Switzerland, Bulletin of the Seismological Society of America, 2021.

Perron, V., Gélis, C., Froment, B., Hollender, F., Bard, P.-Y., Cultrera, G., and Cushing, E. M.: Can broad-band earthquake site responses be predicted by the ambient noise spectral ratio? Insight from observations at two sedimentary basins, Geophysical Journal International, 215, 1442–1454, 2018.

Poggi, V., Edwards, B., and Fäh, D.: Derivation of a reference shear-wave velocity model from empirical site amplification, Bulletin of the705Seismological Society of America, 101, 258–274, 2011.

van Ginkel, J., Ruigrok, E., and Herber, R.: Assessing soil amplifications in Groningen, the Netherlands, First Break, 37, 33–38, 2019.